# Pathobiology and dysbiosis of the respiratory and intestinal microbiota in 14 months old Golden Syrian hamsters infected with SARS-CoV-2

Brittany Seibert[1], C. Joaquín Cáceres[1], Silvia Carnaccini[1], Stivalis Cardenas-Garcia[1], L. Claire Gay[1], Lucia Ortiz[1], Ginger Geiger[1], Daniela S. Rajao[1], Elizabeth Ottesen[2], Daniel R. Perez[1] *

1 Department of Population Health, College of Veterinary Medicine, University of Georgia, Athens, Georgia, United States of America, 2 Department of Microbiology, University of Georgia, Athens, Georgia, United States of America

☯ These authors contributed equally to this work.
* dperez1@uga.edu

**Data Availability Statement:** The 16S sequencing data set was deposited under BioProject PRJNA848775.

## Abstract

The pandemic of severe acute respiratory syndrome coronavirus 2 (SARS2) affected the geriatric population. Among research models, Golden Syrian hamsters (GSH) are one of the most representative to study SARS2 pathogenesis and host responses. However, animal studies that recapitulate the effects of SARS2 in the human geriatric population are lacking. To address this gap, we inoculated 14 months old GSH with a prototypic ancestral strain of SARS2 and studied the effects on virus pathogenesis, virus shedding, and respiratory and gastrointestinal microbiome changes. SARS2 infection led to high vRNA loads in the nasal turbinates (NT), lungs, and trachea as well as higher pulmonary lesions scores later in infection. Dysbiosis throughout SARS2 disease progression was observed in the pulmonary microbial dynamics with the enrichment of opportunistic pathogens (*Haemophilus*, *Fusobacterium*, *Streptococcus*, *Campylobacter*, and *Johnsonella)* and microbes associated with inflammation (*Prevotella*). Changes in the gut microbial community also reflected an increase in multiple genera previously associated with intestinal inflammation and disease (*Helicobacter*, *Mucispirillum*, *Streptococcus*, unclassified Erysipelotrichaceae, and Spirochaetaceae). Influenza A virus (FLUAV) pre-exposure resulted in slightly more pronounced pathology in the NT and lungs early on (3 dpc), and more notable changes in lungs compared to the gut microbiome dynamics. Similarities among aged GSH and the microbiome in critically ill COVID-19 patients, particularly in the lower respiratory tract, suggest that GSHs are a representative model to investigate microbial changes during SARS2 infection. The relationship between the residential microbiome and other confounding factors, such as SARS2 infection, in a widely used animal model, contributes to a better understanding of the complexities associated with the host responses during viral infections.

**Funding:** This study was supported by a subcontract from the Center for Research on Influenza Pathogenesis (CRIP) to DRP under contract HHSN272201400008C from the National Institute of Allergy and Infectious Diseases (NIAID) Centers for Influenza Research and Surveillance (CEIRS). This research was also supported by the University of Georgia College of Veterinary Medicine Office of Research and Faculty and Graduate Affairs (ORFGA) Competitive Research Grant for Graduate Students. DRP receives funds from the Georgia Research Alliance and the Caswell S. Eidson endowment fund, University of Georgia. The funders had no role in study design, data collection and analysis, decision to publish, or preparation of the manuscript.

**Competing interests:** The authors have declared that no competing interests exist.

## Author summary

The SARS-CoV-2 pandemic led to millions of human losses, notably affecting the geriatric population, who are at greater risk of developing acute respiratory distress infection leading to prolonged hospitalization and death. However, the mechanism of age-related pathogenicity is not fully understood. Here, we utilized an aged Syrian hamster model resembling ~60-year-old humans to analyze the pathobiology, host response, and effects of SARS2 on the respiratory and intestinal microbiome. We identified specific microbial markers observed in severe COVID-19 patients within the lungs of aged hamsters infected with SARS-CoV-2. Prior influenza A virus (H1N1) exposure amplified these changes. Similarities among aged GSH and critically ill COVID-19 patients suggest that GSHs are a valuable model for investigating microbial changes during SARS2 infection. The relationship between the age, residential microbiome and viral pathogens contributes to a better understanding of the complexities associated with the host responses during viral infection while limiting potential environmental factors that may contribute to inter-individual variation.

## Introduction

The severe acute respiratory syndrome coronavirus 2 (SARS2) is responsible for the ongoing coronavirus disease 2019 (COVID-19) pandemic. Clinical manifestations of infection are highly variable, ranging from asymptomatic or mild nonspecific flu-like symptoms to severe symptoms such as pneumonia, acute respiratory distress syndrome, multiple organ failure, and death [1,2]. Several risk factors have been identified for severe COVID-19 disease, including sex, age, obesity, cardiovascular conditions such as hypertension, smoking, among others [1,2]. Older individuals are at greater risk of developing acute respiratory distress amid SARS2 infection [3], leading to prolonged hospitalization and death [2]. Animal models resembling human disease have been essential in understanding SARS2 pathogenesis. Golden Syrian hamsters (GSH) are naturally susceptible to SARS2 replication in the respiratory tract, inducing severe lung pathology [4,5]. Generally, GSH at 5–8 weeks of age have been used in experimental studies [6–11]; however, multiple groups reported that 8-10-month-old GSHs develop more severe disease by exhibiting increased weight loss and slower lung recovery post-infection, emulating more severe disease presentation in older individuals [12–14]. The vulnerability of enhanced illness in the elderly population has been hypothesized to encompass a combination of immunopathology from exacerbated immune response and immunosenescence [2]. However, the mechanism and characteristics of age-related changes to the host such as the relationship between the immune response and host residential microbes during respiratory infections warrants further investigation.

While relatively established during adulthood, the residential host microbiota undergoes significant shifts in composition and diversity during the aging process, which can directly or indirectly affect the regulation of the immune system [15,16]. The deep respiratory microbiome is characterized by a diverse bacterial community in low abundances [17]. In healthy individuals, the lung microbiota is generally dominated by commensal bacteria within the phyla Firmicutes such as *Veillonella spp* and Bacteroidetes such as *Prevotella spp* [17]. Influenza A virus (FLUAV) and SARS2 infection affect the local bacterial community through abundance and composition changes [18–21]. Previous studies in humans reported that the lung microbiota of SARS2 infected patients consisted of opportunistic pathogens or enrichment of commensal bacteria such as *Pseudomonas*, Enterobacteriaceae, and *Acinetobacter* [18, 20].

Respiratory viral infections can also affect and induce changes in the gut microbiota through the gut-lung axis [22,23]. Further, gastrointestinal symptoms along with viral RNA (vRNA) in the feces have been observed occasionally during SARS2 infection [24–26]. While the direct mechanism is unclear, intestinal dysbiosis in COVID-19 patients has been previously described in various human cohort studies [21,27,28]. COVID-19 patients with severe symptoms presented dysbiosis of the gut microbiota characterized by commensal bacteria depletion and pathogenic bacteria enrichment [21,28–30]. Overall, fecal microbiome studies reported an association of opportunistic pathogens with COVID-19 infection, including *Streptococcus*, *Rothia*, *Veillonella*, *Erysipelatoclostridium*, *Actinomyces*, *Collinsella*, and *Morganella* [21,29]. While not fully understood, the intestinal microbiota affects various physiological functions, including metabolism, digestion, organ function, response to invading pathogens, and immune homeostasis [31]. While human clinical samples are informative, there are also limitations, such as the environmental factors influenced by daily life (age, geographic location, diet, sex) or clinical management (antiviral or antibiotic treatment) that contribute to inter-individual variation. Human microbiome studies are limited since samples are mostly restricted to those from the nasopharynx or feces; lungs and intestinal tissues are more difficult to collect. However, it is well accepted that microbial communities vary in composition and abundance depending on different parts of the respiratory and intestinal tracts. Therefore, it is valuable to characterize the effect of SARS2 infection on the respiratory and intestinal microbiota in established animal models.

We recently analyzed the lung and cecum microbiota of K18-hACE2 mice after a low and high dose SARS2 infection [32]. Within this animal model, we reported diversity and compositional changes in the respiratory and intestinal microbiota in a virus dose-dependent manner [32]. However, the artificial expression of hACE2 does not mirror the natural susceptibility observed in humans. Therefore, we utilized an aged Syrian hamster model to analyze the pathobiology, host response and effects of SARS2 and pre-exposure of FLUAV H1N1 on the respiratory and intestinal microbiome in older patients. SARS2 challenged GSHs with and without prior FLUAV exposure had similar vRNA loads for respiratory and non-respiratory tissues except that FLUAV pre-exposure led to decreased SARS2 vRNA in the heart and nasal turbinates (NT). While SARS2 challenge led to overall higher pulmonary lesion scores later (6 days post challenge (dpc)) in infection than in the FLUAV pre-exposure group, FLUAV pre-exposed GSHs displayed slightly more pronounced pathology in the NT and lungs at 3 dpc. SARS2 challenge also impacted the microbial dynamics by enriching common opportunistic pathogens such as *Haemophilus*, *Fusobacterium*, *Streptococcus*, *Campylobacter*, and *Johnsonella*. Conjointly, SARS2 challenge led to dysbiosis in the small intestine (SI), cecum, and feces associated with more pronounced lesions in the cecum, notably in GSHs with FLUAV exposure. Several bacterial taxa were enriched in the SARS2 challenged GSH including *Helicobacter*, *Mucispirillum*, *Ileibacterium*, *Streptococcus*, unclassified Erysipelotrichaceae, Eubacteriaceae, and Spirochaetaceae while *Allobaculum* and unclassified Lachnospiraceae were depleted. Multiple taxa also correlated with vRNA load and pathological scores in respiratory and non-respiratory tissues. Altogether, SARS2 challenge with and without FLUAV exposure presented distinct disease pathology while both altered the respiratory and intestinal microbiota, suggesting that older GSH may be paramount in further investigating the effect of age on the host response to SARS2 infection.

## Results

### SARS2 infection in 14 months old GSH led to high viral loads, enhanced pathology, and consistent weight loss

GSHs were divided into 3 groups: SARS2 GSHs were challenged intranasally with 1 x 10$^5$ TCID50/hamster of SARS2 while FLUAV-SARS2 GSHs were pre-exposed to FLUAV (primed

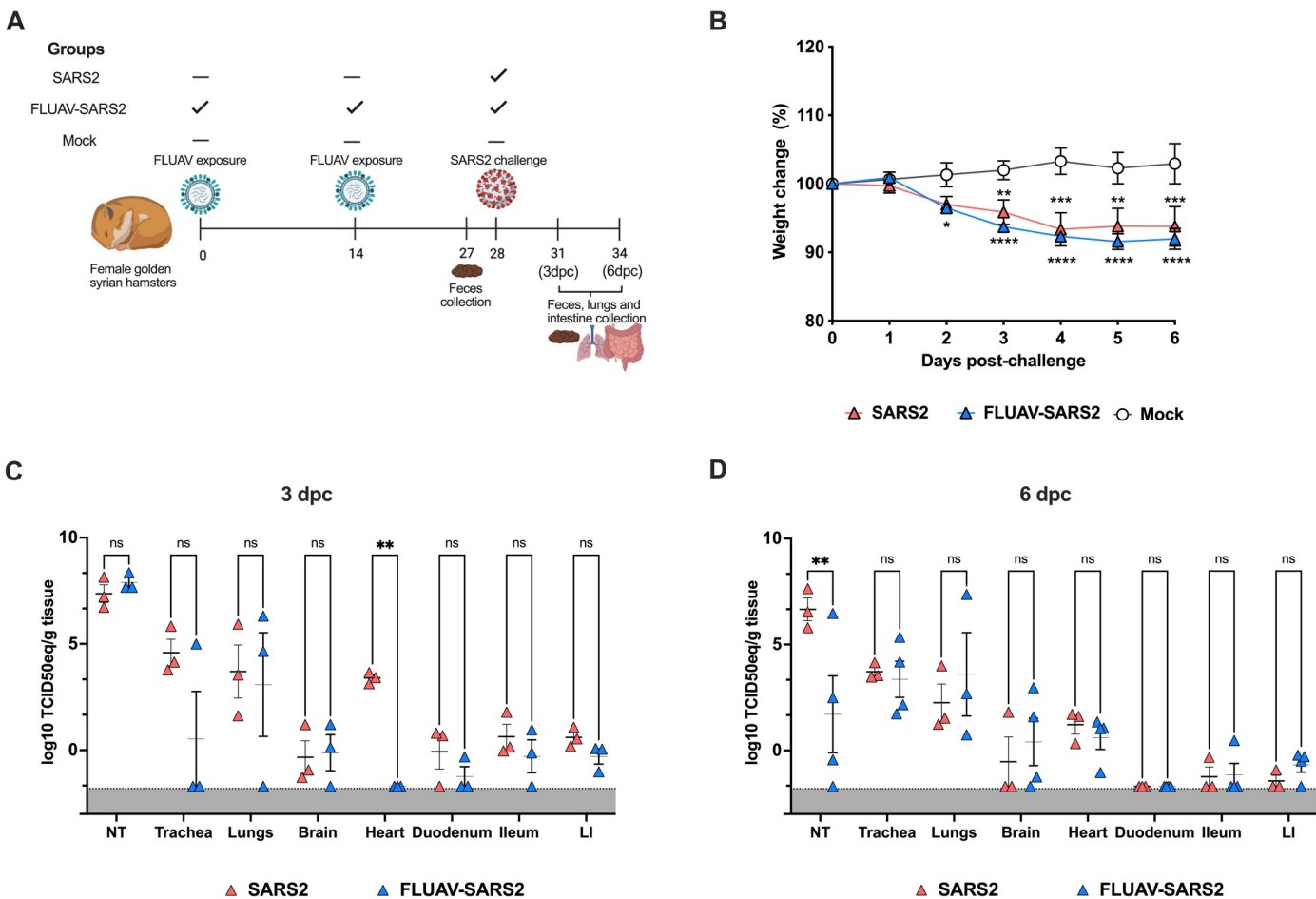

**Fig 1. Experimental design, body weight, and viral loads of aged GSHs post-SARS2 challenge.** (A) 14-month female GSHs were randomly distributed into 3 groups: SARS2 GSHs were challenged with SARS2 (n = 6), FLUAV-SARS2 GSHs were pre-exposed to FLUAV twice and then challenged with SARS2 (n = 7), and mock GSH remained unchallenged (n = 3). A subset of GSH (n = 3 group/time point) were humanely euthanized at 3 and 6 dpc, and numerous tissues were collected. Feces was collected 1 day before challenge, and then 3 and 6 dpc for the remaining individuals. Illustration created with BioRender.com. (B) Weight change was calculated for all groups up to 6 dpc, including SARS2 (red triangles), FLUAV-SARS2 (blue triangles), and mock (white circles). Viral loads at (C) 3 dpc and (D) 6 dpc from the challenged groups were determined by RT-qPCR and conveyed as log10 TCID50 equivalent (TCID50eq) per gram of tissue. Error bars represent the mean ± SEM. Statistical analysis was performed by two-way ANOVA with a Dunnett's multiple comparisons test, * <0.05. NT- nasal turbinates, LI–large intestine.

twice, 14 days apart at 1 x $10^5$ TCID50/hamster) 13 days before SARS2 challenge (as above). FLUAV exposure was used to determine if such treatment would affect SARS2 infection, disease progression, and microbiome composition outcomes. The mock GSH group was challenged with 1XPBS. Clinical signs, including weight loss, SARS2 shedding and histopathological changes of different tissues were analyzed at 3 and 6 dpc (Fig 1A). Assessment of humoral responses against FLUAV obtained on 1 day prior to SARS2 challenge (27 days post $1^{st}$ FLUAV dose) revealed antibody responses, hemagglutination inhibition (HAI) and enzyme-linked immunosorbent assay (ELISA) below the limit of detection suggesting that inoculation of FLUAV in GSH led to restricted and/or abortive FLUAV replication. Due to the limited number of ~14-month-old hamsters, active FLUAV replication was not evaluated within this study. The effects, if any, of prior-FLUAV exposure in GSH challenged with SARS2 would likely not be related to stimulation of antibody responses. After SARS2 challenge, the GSHs experienced clinical signs consistent with virus infection regardless of prior-FLUAV exposure starting at 2 dpc (SARS2: range 1.15–6.5% and FLUAV-SARS2: range 1.19–7.4%)

and continued to deteriorate until 6 dpc when the study was terminated (SARS2: range 7.7–10.2% and FLUAV-SARS2: range 4.3–11.7%) (Fig 1B). Overall, a significant weight loss was observed upon SARS2 challenge regardless of previous FLUAV exposure.

In addition to weight loss, high SARS2 vRNA loads were detected from the NT, trachea, and lungs in both the SARS2 and FLUAV-SARS2 groups with no statistically significant differences at 3 and 6 dpc (Fig 1C and 1D). The exception was on vRNA load data from NT at 6 dpc, where statistically significant higher viral loads were detected in the SARS2 group (p = 0.0038; Two-way ANOVA). Further, 2 out of 3 trachea samples and 1 out of 3 lung samples had vRNA loads below limit of detection in the FLUAV-SARS group at 6 dpc. Notably at 3 dpc, vRNA loads were detected in the heart from the SARS2 group, but not from the FLUAV-SARS2 group (p = 0.0096; Two-Way ANOVA). Low to undetectable vRNA loads were observed in the brain and intestine (duodenum, ileum, and cecum) at both 3 and 6 dpc in both the SARS2 and FLUAV-SARS2 challenged groups. Collectively, SARS2 challenge resulted in elevated vRNA loads in respiratory tissues and heart while FLUAV pre-exposure led to reduced SARS2 vRNA within the NT (at 6 dpc) and the heart (at 3 dpc). These observations suggest that prior FLUAV exposure led to a potential antiviral state that limited and/or delayed SARS2 replication in aged GSHs.

Histopathological analysis of GSH groups revealed that the organs most affected upon infection were the NT, lungs, tracheas, and large intestine. Overall, lesions appeared slightly more pronounced in the FLUAV-SARS2 GSH, with a few exceptions (Table 1). In both SARS2 challenged groups, lesions in the NT were more prominent at 3 dpc (Fig 2A and 2B) than at 6 dpc (Fig 2D and 2E). These ranged from mild-moderate to moderate-severe, multifocal, acute-subacute suppurative rhinitis with rare syncytia formation and abundant intraluminal catarrhal to fibrinosuppurative exudate deposition. Within both groups, variable amounts of intracytoplasmic virus antigen immunolabeling was present in epithelial cells (S1A–S1F Fig).

Tracheas presented minimal variations of findings between challenged groups at given time points. Scores were higher at 3 dpc (Fig 2G and 2H) versus at 6 dpc (Fig 2J and 2K), and lesions consisted of mild to moderate, multifocal, acute-subacute neutrophilic and histiocytic tracheitis with intraluminal muco-catarrhal exudate formation. No syncytia were found in any of these sections.

At 3 dpc, pulmonary lesions were overall milder in SARS2 GSH than in FLUAV-SARS2 GSH (Fig 2M and 2N and Table 1). In both SARS2 challenged groups, lesions were centered on the bronchi and bronchioles. They predominantly consisted of mild-moderate (Fig 2M) and moderate-severe (Fig 2N) multifocal, acute suppurative bronchitis/bronchiolitis with intraluminal exudate deposition. Syncytia were rarely observed in the bronchial respiratory epithelium at 3 dpc. At this time, peribronchial inflammation was mild, whereas perivascular cuffing and pulmonary vascular endothelialitis were accompanied by rare intravascular thrombi. Further, moderate amounts of virus antigen were present in bronchiolar epithelium

**Table 1. H&E pathology scores of respiratory tissues.** Subjectively scored tissues (blind) based on percentage of the total parenchyma affected by lesions and inflammation as: none (0%), mild; <15% (1), mild to moderate; 16–30% (2), moderate; 31–50% (3), moderate to severe 51–75% (4) and severe; ≥75% (5). A dash is used to separate the score evaluated for each individual hamster separated by group and days post challenge (dpc). NA-not applicable.

| Group | dpc | Nasal Turbinates | Trachea | Lung |
|---|---|---|---|---|
| SARS2 | 3 | 3 / 3 / 3 | 1 / 2 / 3 | 2 / 2 / 2 |
|  | 6 | 3 / 2 / 3 | 1 / NA / 1 | 5 / 4 / 5 |
| FLUAV-SARS2 | 3 | 3 / 4 / 4 | 1 / 3 / 1 | 3 / 4 / 3 |
|  | 6 | 3 / 2 / 3 / 3 | 2 / 1 / NA / NA | 3 / 4 / 5 / 4 |
| Mock | 3 | 0 | 0 | 0 |
|  | 6 | 0 / 0 | 0 / 0 | 0 / 0 |

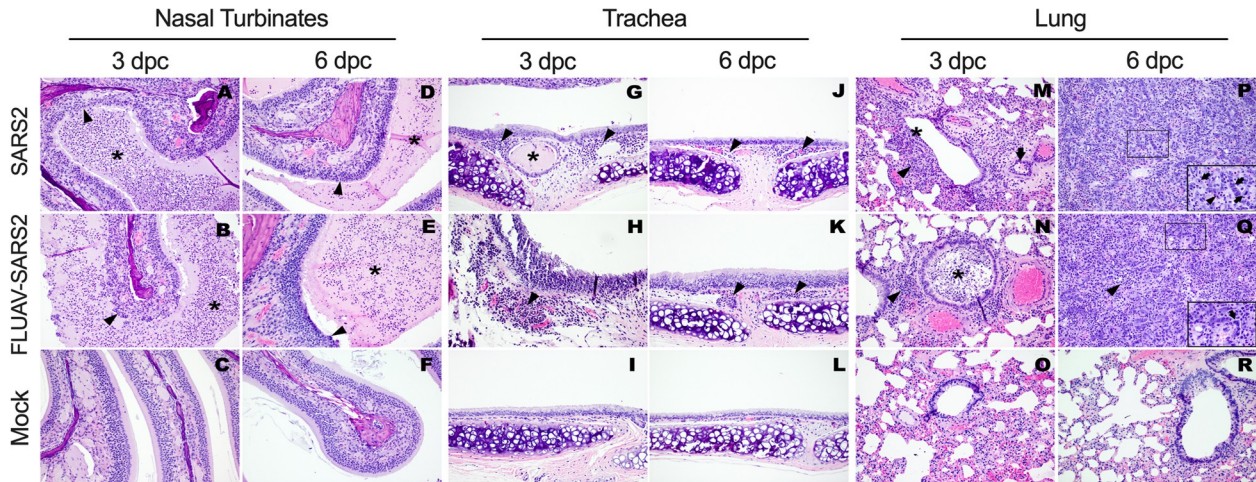

**Fig 2. Histopathologic findings in the different respiratory tissues of SARS2, FLUAV-SARS2, and mock-challenged hamsters at given timepoints.** (A-F) Nasal turbinates. (G-L) Tracheas. (M-R) Lungs. (A-B) Moderate to severe necrotizing rhinitis in the infected hamsters at 3 dpc. The surface epithelium is necrotic, infiltrated by neutrophils and sloughing into the lumen (arrowhead). The lumen is filled with abundant intraluminal suppurative exudate (asterisk). (D-E) Rhinitis is still present in the infected hamsters at 6 dpc. The epithelium is irregular, with fewer necrotic cells sloughing into the lumen (arrowheads). The lumen is filled with abundant intraluminal suppurative exudate (asterisk). (C, F) No significant findings are observed in mock hamsters' nasal turbinates. (G-H) Clusters of neutrophils and lymphocytes are expanding the mucosal lamina propria (arrowheads) of infected hamsters' tracheas at 3 dpc. Occasionally, tracheal glands are dilated by intraluminal eosinophilic material (asterisk). (J-K) Similar small mix of neutrophils, macrophages, and lymphocytes (arrowheads) are observed in the tracheas of infected hamsters at 6 dpc. (I, L) No significant findings are observed in mock hamsters' tracheas. (M-N) At 3 dpc, the bronchiolar mucosa in infected hamsters is necrotic and sloughing into the lumen (asterisk). Small to moderate clusters of neutrophils and lymphocytes are infiltrating the peribronchiolar interstitium and bronchiolar mucosa (arrowheads). (P-Q) At 6-dpc, pneumonia is characterized by florid pneumocyte type II hyperplasia following the outline of the alveoli (arrows within insert: bottom right corner; HE, 40X), and syncytia. Moderate to large numbers of neutrophils, macrophages and lymphocytes are expanding the septa and pouring into the alveoli (arrowheads). (O, R) No significant findings are observed in the mock hamsters' lungs. All H&E images are pictured at 20X magnification.

and intraluminal necrotic cellular debris (S1G and S1H Fig). At 6 dpc more severe pulmonary lesion scores were observed in the SARS2 group than in FLUAV-SARS2 group. These consisted of moderate to severe, multifocal to diffuse, subacute interstitial pneumonia with marked alveolar proliferation of cuboidal epithelial cells (pneumocytes type II), replacing the necrotic pneumocytes type I (Fig 2P and 2Q). Pneumocytes type II often presented karyomegaly and multinucleation (syncytia). Alveoli were often collapsed or filled with hemorrhagic and suppurative exudate. Moderate multifocal, acute fibrinosuppurative pleuritis was observed mostly in the FLUAV-SARS2 group at 6 dpc. However, intracytoplasmic virus antigen staining was scarce at 6 dpc compared to 3 dpc (S1J and S1K Fig).

In the sections of small intestines, the inflammation was generally minimal and present in all groups, including the mock challenged, at both timepoints. Only in the SARS2 challenged groups, small numbers of eosinophils were migrating to the enterocyte lining (S2A–S2E Fig) and associated with mild non-specific lesions to the villi. Variable numbers of intraluminal protozoa and mixed bacteria appeared subjectively increased in the challenged GSHs. Protozoal morphologic features were mostly compatible with *Giardia muris* (S2E Fig–insert).

Typhlocolitis was more pronounced within the large intestine in the FLUAV-SARS2 GSH than in SARS2 GSH at both timepoints. Enterocytes were degenerating, sloughing into the lumen or within the glands, and were colonized by moderate numbers of mixed coccobacilli and filamentous bacteria. Especially in the challenged groups, there was a visible increase in *Entamoeba muris* and bacteria numbers (S2G, S2I and S2L Fig–inserts).

Findings in the brain were inconsistent. Only 2 GSHs presented evidence of mild suppurative meningitis, one in SARS2 group and one in the FLUAV-SARS2 group at different time

points (S1 Table). No significant findings were observed in the heart sections in any group and timepoints. These results suggest that in the absence of FLUAV-pre-exposure, increased pulmonary lesions were seen in the later part of infection after SARS2 challenge. In contrast, FLUAV pre-exposure led to exacerbated pathological scores in the NT and lungs early after SARS2 challenge. FLUAV pre-exposure also led to more pronounced lesions in the cecum throughout infection after SARS2 challenge.

## SARS2 challenge and FLUAV pre-exposure alters the composition of the lung microbiota in aged GSHs

To characterize the lung microbiome, we sequenced the V4 region of 16S rRNA genes amplified from lung homogenates collected at 3 and 6 dpc. Alpha diversity (observed number of amplicon sequence variants (ASV's)) and Shannon diversity showed a trend towards an increased number of ASVs and diversity following SARS2 challenge, particularly in the FLUAV-SARS2 group; however, no significant differences were observed among the SARS2 challenged groups and mock when dpc were combined (Fig 3A). To determine changes in the lung microbial composition after SARS2 challenge, we analyzed beta diversity using unweighted Jaccard and weighted Bray-Curtis's dissimilarity metrics through non-metric multidimensional scaling (NMDS) ordination. Both ordinations showed clustering by group (R2 = 0.22, p = 0.0030; PERMANOVA) and dpc (R2 = 0.096, p = 0.025; PERMANOVA), though the interaction between the terms was not significant (Figs 3B and S3A). The dissimilarity distance between the two SARS2 challenged groups and the mock was larger than the distance between the two challenged groups (FLUAV-SARS2: p < 0.0001 and SARS2: p = 0.00027; Wilcox) (Fig 3C). When analyzing the dispersion within the treatment groups using the unweighted and weighted distances, both groups challenged with SARS2 had lower intra-group variation than the mock (FLUAV-SARS2: p = 0.015 and SARS2: p = 0.037; Wilcox); however, the difference in sample size could explain the results (Figs 3C and S3B). Overall, SARS2 challenge impacts the diversity within the lung microbial dynamics through changes in the relative abundance of different microbes, while prior FLUAV exposure amplified these changes.

At the phylum level, the lung microbiota was dominated by Bacteroidota (Bacteroidetes) (mean ($\bar{x}$): 38.68%) and Firmicutes ($\bar{x}$: 32.50%) in all three groups when both dpc were combined (Fig 3D). Prominently, the abundance of Fusobacteriota ($\bar{x}$: SARS2: 6.57%, FLUAV-SARS2: 6.98%, mock: 0.60%) and Proteobacteria ($\bar{x}$: SARS2: 25.79%, FLUAV-SARS2: 22.54%, mock: 2.6%) increased in the SARS2 challenged GSHs plausibly at the expense of Firmicutes ($\bar{x}$: SARS2: 23.76%, FLUAV-SARS2: 28.87%, mock: 58.44%). A difference in relative abundance was also observed at lower taxonomic levels. Strikingly, a large increase in the relative abundance of *Haemophilus* was observed at 3 dpc ($\bar{x}$: SARS2: 40.96%, FLUAV-SARS2: 24.74%, mock: 0%) and 6 dpc ($\bar{x}$: SARS2: 15.49%, FLUAV-SARS2: 15.23%, mock: 0%) in both SARS2 challenged groups, consistent with overall increase of Proteobacteria (Fig 3E). While the cumulative *Haemophilus* relative abundance was similar at 6 dpc among the challenged groups, a higher prevalence of the genera was observed in FLUAV-SARS2 GSH (4/4) compared to SARS2 GSH (1/3). Increased relative abundances of *Prevotella 7* ($\bar{x}$: SARS2: 5.93%, FLUAV-SARS2: 4.14%, mock: 0%), *Streptococcus* ($\bar{x}$: SARS2: 5.85%, FLUAV-SARS2: 2.24%, mock: 0%), *Johnsonella* ($\bar{x}$: SARS2: 1.45%, FLUAV-SARS2: 2.01%, mock: 0%), and *Fusobacterium* ($\bar{x}$: SARS2: 8.54%, FLUAV-SARS2: 9.31%, mock: 0%) were also observed in both SARS2 challenged groups compared to the mock (Fig 3E).

We performed differential abundance testing at the genus level using three programs, DeSeq2, linear discriminant analysis effect size (LefSE), and ALDEx2. At the genus level,

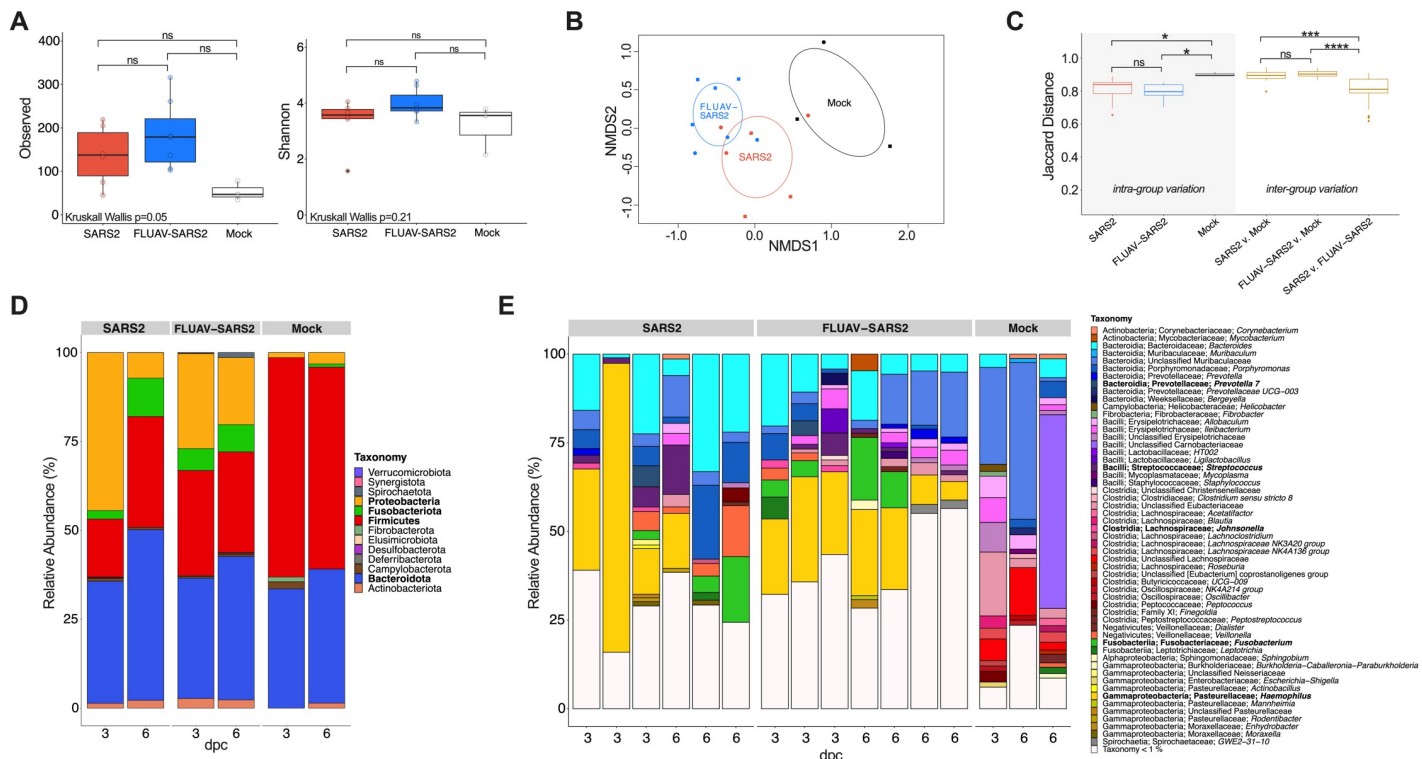

**Fig 3. Diversity and taxonomic abundances of the lungs in SARS2, FLUAV-SARS2 and mock-challenged GSH.** (A) Alpha diversity measure of the observed number of ASVs (left) and Shannon diversity (right) between SARS2 (red), FLUAV-SARS2 (blue), and mock (white) containing all dpc from the rarified ASV count table. Multiple group comparisons were performed using Kruskal-Wallis, while pair-wise comparisons were conducted using the Wilcox test with Bonferroni correction. (B) NMDS plot of unweighted Jaccard dissimilarity distance. The color designates groups (SARS2: red, FLUAV-SARS2: blue, and mock: black), and dpc is categorized by the shape (3 dpc: circle and 6 dpc: square). Ellipses were constructed using the standard deviation. (C) Comparison of unweighted Jaccard dissimilarity distance within each group and across multiple comparisons. Intra-group variation is marked with grey shading, while the inter-group comparisons are marked gold. Pair-wise comparisons were performed using the Wilcox test with Bonferroni correction. (D) Relative abundances agglomerated at the phylum level separated by group and dpc. Phyla in bold are those discussed within the results. (E) Relative abundances of each hamster at the lowest taxonomic rank identified separated by group and ordered by dpc. Taxa in bold are those discussed within the results. ns: non-significant (p > 0.05); * <0.05, ** < 0.005, *** < 0.0005, **** < 0.00005.

*Campylobacter*, *Fusobacterium*, *Haemophilus*, and *Moraxella* were identified as significantly enriched within the SARS2 group compared to the mock controls by 2 of 3 (Deseq2 and LefSE) differential analyses (Fig 4A and S2 Table). *Prevotella 7*, *Johnsonella*, *Streptococcus*, *Fusobacterium*, and *Haemophilus* were identified as significantly enriched within the FLUAV--SARS2 GSHs in comparison with mock-challenge animals (Fig 4B and S2 Table). Other taxa were considered significant in each group but were only identified in 1 of 3 differential analyses (S4 Fig). In contrast, no genera were considered significantly different in at least 2/3 differential analysis when comparing SARS2 and FLUAV-SARS2 GSHs (S2 Table and S4E–S4G Fig). Collectively, SARS2 challenge with and without prior FLUAV exposure impacts the microbiota by enriching opportunistic pathogens; however, previous FLUAV exposure partially modulates this phenomenon with enrichment of *Haemophilus* later (6 dpc) in SARS2 infection.

## Altered composition of the lung microbiota correlates with SARS2 vRNA loads and pathology

We examined the association of multiple genera and unclassified families to SARS2 vRNA loads and pathology scores using Spearman correlation. *Prevotella 7*, *Fusobacterium*, and *Haemophilus* positively correlated with SARS2 challenge while *Prevotellaceae UCG-003* and

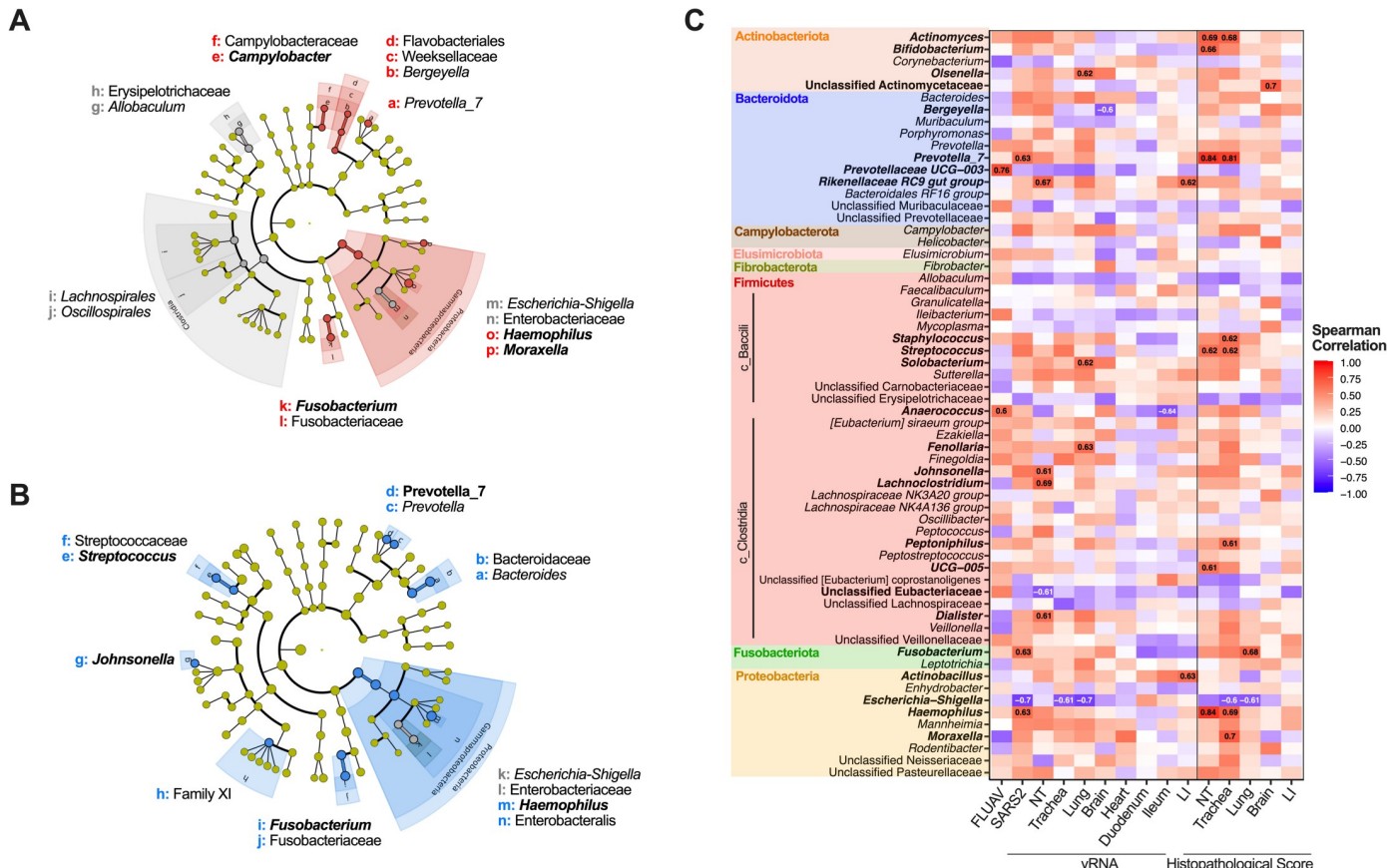

**Fig 4. Differential taxonomic analysis of the lung microbiota in SARS2, FLUAV-SARS2 and mock-challenged GSH.** (A-B) Cladogram of significantly (p<0.05) abundant taxa between (A) SARS2 (red) and mock (grey) or (B) FLUAV-SARS2 (blue) and mock (grey) using LEfSe analysis with default parameters. Taxa displayed in bold were significant in at least 2/3 differential analyses performed (Deseq2, ALDEx2, and LefSE). (C) A heatmap illustrating the spearman correlation among different infection factors and the relative abundance of bacterial taxa within the lungs. Correlations among pathology of the heart and SI were removed since the pathological score was the same for all GSH. Correlation labels and taxa displayed in bold have a spearman correlation value > 0.60 or < -0.60. SARS2 – challenged with SARS2, FLUAV–inoculated with FLUAV and then challenged with SARS2 (mock-challenged GSH were not included), NT- nasal turbinates, LI– large intestine.

*Anaerococcus* positively correlated with FLUAV pre-exposure (Fig 4C). *Escherichia-Shigella* negatively correlated with SARS2 challenge. We also observed a strong correlation between several bacterial taxa and SARS2 vRNA load in tissues, particularly within the respiratory tract (Fig 4C). For instance, *Rikenellaceae RC9 gut group*, *Johnsonella*, *Lachnoclostridium*, and *Dialister* positively correlated with increased vRNA in the NT while *Olsonella*, *Solobacterium*, and *Fenollaria* positively correlated with increased vRNA in the lungs. Within the trachea and the lungs, *Escherichia-Shigella* negatively correlated with vRNA loads. In non-respiratory tissues, *Moraxella* positively correlated with increased vRNA in the heart, and *Rikenellaceae RC9 gut group* and *Actinobacillus* positively correlated with increased vRNA in the cecum. On the other hand, *Anaerococcus* negatively correlated with vRNA in the ileum.

Strong correlations were also observed between several bacterial taxa and histopathological scores (Fig 4C and Tables 1 and S1). *Prevotella 7* and *Haemophilus* positively correlated with more severe pathology scores in the NT and trachea. *Actinomyces*, *Bifidobacterium*, *Streptococcus*, and *UCG-005* positively correlated with more severe pathology scores in the NT, and *Actinomyces*, *Staphylococcus*, *Peptoniphilus*, and *Moraxella* in the trachea. Within the lungs, *Fusobacterium* positively correlated with more severe pathology while *Escherichia-Shigella*

negatively correlated with increased severity in the trachea and the lungs. In non-respiratory tissues, unclassified Actinomycetaceae positively correlated with more severe pathology in the brain, while no genera were found to correlate with severity of the cecum. Correlations among the heart and SI pathology were not determined since the pathological score was the same for all GSH groups (S1 Table).

## SARS2 challenge and FLUAV pre-exposure alters the intestine and fecal microbiota composition in aged GSHs

The V4 region of 16S rRNA genes amplified from the SI (duodenum and ileum), large intestine (cecum), and feces collected at 3 and 6 dpc was sequenced and analyzed. Fecal samples from 1 day before challenge (-1 dpc) were analyzed as a separate pre-challenge group (Fig 1A). Rarefaction curves of alpha diversity (observed number of ASV) and Shannon diversity showed adequate coverage and no significant differences among the groups when both dpc were combined; however, a difference in observed species among the sample sites was apparent (Figs 5A, 5B and S5A). The cecum had the highest number of ASVs, followed by feces, ileum, and duodenum (Fig 5B). When analyzing the effect of FLUAV exposure on pre-challenge fecal samples, no significant alpha diversity differences were observed (S5B Fig). For

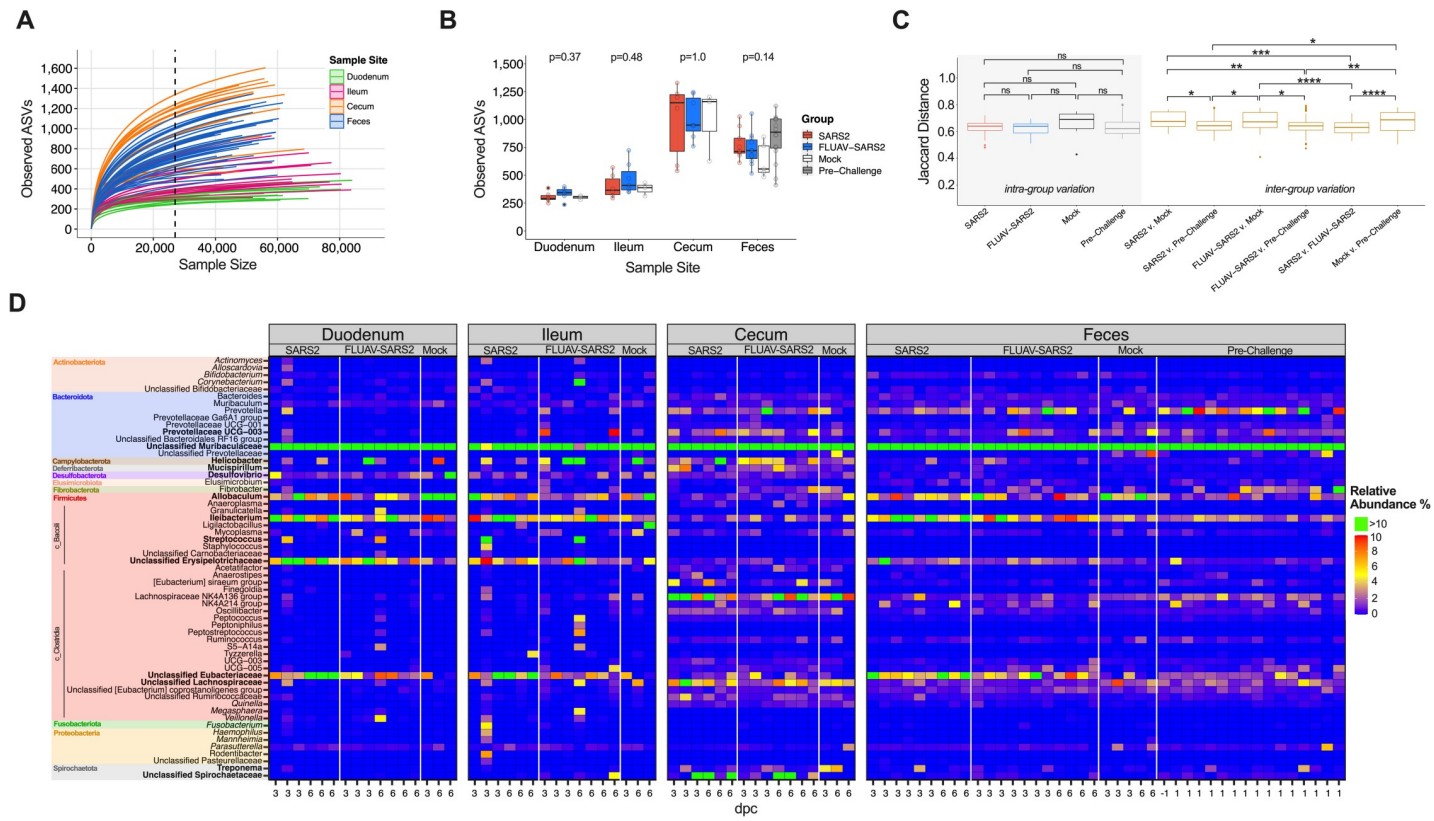

**Fig 5. Diversity and taxonomic abundances of the intestine and feces in SARS2, FLUAV-SARS2 and mock-challenged hamsters.** (A) Rarefaction curve of the intestine and fecal samples colored by sample type. The dashed line (x = 27,020) represents the number of reads each sample was rarified for downstream diversity analyses. (B) The observed number of ASVs between FLUAV-SARS2 (red), SARS2 (blue), mock (white), and pre-challenge (grey) containing all dpc of different sample types from the rarified ASV count table. Statistical analysis comparing the different groups within each sample type was performed using Kruskal-Wallis. (C) Comparison of unweighted Jaccard dissimilarity distance within each group and across multiple comparisons in feces. Intra-group variation is marked with grey shading, while the inter-group comparisons are colored gold. Pair-wise comparisons were performed using the Wilcox test with Bonferroni correction. Comparisons that are not indicated by an asterisk were considered non-significant (p > 0.05). (D) Heatmap of the relative abundances of taxa with a relative abundance > 1%. Relative abundances greater than 10% are shown in green. Taxa are color-coded by phylum. Taxa in bold are those discussed within the results. * <0.05, ** < 0.005, *** < 0.0005, **** < 0.00005.

downstream analysis, the duodenum and ileum samples were combined to encompass the SI. Beta diversity using NMDS ordination plots showed limited clustering of groups in unweighted (Jaccard) and weighted (Bray-Curtis) dissimilarity metrics (S6 Fig); however, PERMANOVA analysis showed significant differences by group in the SI (R2 = 0.12, p<0.001) and feces (R2 = 0.14, p<0.001) but not within the cecum. There was also considerable overlap among the pre-exposed FLUAV GSH and mock GSH when analyzing the pre-challenge fecal samples (S6G-H). When analyzing the inter-variation among the groups, the dissimilarity distance was similar within the SI and the cecum (S7A and S7B Fig). The SARS2 GSHs had lower intra-group variation than the mock challenged GSHs in the feces but not considered statistically significant (Fig 5C). Also in the feces, the dissimilarity distance comparing both SARS2 challenged groups to the mock was larger than the distance between the comparison of SARS2 versus FLUAV-SARS2 (p < 0.001; Wilcox) (Fig 5C). The distance between SARS2 challenged GSH and pre-challenge was not significantly different from the distance between SARS2 and FLUAV-SARS2 GSH (p = 1; Wilcox) but was significantly different from the comparisons with the mock (SARS2: p = 0.0033 and FLUAV-SARS2: p = 0.0024; Wilcox). These findings suggest that the fecal microbiome composition of aged GSH challenged with SARS2 differs from the mock challenged GSH; however, the difference detected among the groups could be partially explained by intra-individual variations observed in the pre-challenge fecal samples.

At the phylum level, the intestinal and fecal microbiota were dominated by Bacteroidota ($\bar{x}$: 44–74%) and Firmicutes ($\bar{x}$: 20–40%) in all groups and sample sites (S7C Fig). However, relative abundance differences were observed at lower taxonomic levels among the different sample sites. The duodenum and ileum shared similar taxonomic profiles that were distinct from the cecum and fecal samples; however, unclassified Muribaculaceae ($\bar{x}$: 56.69%) was the most prominent family among sample sites and groups when both dpc were combined (Figs 5D and S7D). Markedly, decreased abundance of *Allobaculum* within both SARS2 groups was observed in the SI ($\bar{x}$: SARS2: 6.72%, FLUAV-SARS2: 3.19%, mock: 12.88%), cecum ($\bar{x}$: SARS2: 0.70%, FLUAV-SARS2: 0.61%, mock: 2.8%), and feces ($\bar{x}$: SARS2: 4.50%, FLUAV-SARS2: 4.23%, mock: 7.36%). Within the SI, *Streptococcus* ($\bar{x}$: SARS2: 2.42%, FLUAV-SARS2: 1.63%, mock: 0.0040%) and unclassified Erysipelotrichaceae, notably at 3 dpc, ($\bar{x}$: SARS2 3 dpc: 9.704%, 6 dpc: 6.58%; FLUAV-SARS2 3 dpc: 8.04%, 6 dpc: 4.88%; mock 3 dpc: 3.20%, 6 dpc: 2.0%) were enriched while *Desulfovibrio* ($\bar{x}$: SARS2: 1.39%, FLUAV-SARS2: 1.08%, mock: 3.30%) was reduced within both groups of SARS2 challenged GSH. Taxa enriched within the cecum in SARS2 challenged GSH included *Prevotellaceae UCG-003* (x: SARS2: 3.03%, FLUAV-SARS2: 2.67%, mock: 0.11%), *Helicobacter* ($\bar{x}$: SARS2: 3.47%, FLUAV-SARS2: 4.03%, mock: 1.81%), *Mucispirillum* ($\bar{x}$: SARS2: 2.46%, FLUAV-SARS2: 2.18%, mock: 0.55%) and unclassified Spirochaetaceae ($\bar{x}$: SARS2: 6.42%, FLUAV-SARS2: 4.56%, mock: 0.02%) while *Treponema* ($\bar{x}$: SARS2: 0.068%, FLUAV-SARS2: 0.065%, mock: 3.76%) was decreased in both challenged groups compared to the mock. Within the feces, *Ileibacterium* ($\bar{x}$: SARS2: 7.55%, FLUAV-SARS2: 6.47%, mock: 3.02%, Pre-challenge: 2.46%), unclassified Erysipelotrichaceae ($\bar{x}$: SARS2: 3.14%, FLUAV-SARS2: 1.92%, mock: 0.86%, Pre-challenge: 0.88%), and unclassified Eubacteriaceae ($\bar{x}$: SARS2: 7.13%, FLUAV-SARS2: 4.64%, mock: 0.94%, Pre-challenge: 1.66%) were enriched while unclassified Lachnospiraceae ($\bar{x}$: SARS2: 1.04%, FLUAV-SARS2: 0.75%, Pock: 3.53%, Pre-challenge: 1.89%) was reduced in both challenged groups compared to the mock.

Multiple taxa were identified as differentially enriched or reduced within the different groups in at least 2 out of 3 differential analyses when sample types were combined (S3 Table and S8 Fig). Genera including *Prevotella*, *Fibrobacter*, and unclassified Spirochaetaceae within the SI and *UCG-005* and unclassified Spirochaetaceae within the cecum were differentially enriched in the FLUAV-SARS2 group versus mock challenge GSH (S4 Table). No genera were considered differentially enriched in the SARS2 group when compared to the mock within the

SI and cecum. On the other hand, *Allobaculum*, unclassified Prevotellaceae and *Mycoplasma* within the SI and unclassified Prevotellaceae and *Allobaculum* within the cecum were differentially enriched in the mock challenged GSH compared to both SARS2 challenged groups. While comparing SARS2 GSH and FLUAV-SARS2 GSH, *Fibrobacter*, unclassified [Eubacterium] coprostanoligenes group, and unclassified Spirochaetaceae within the SI and *Fibrobacter* and *UCG-005* within the cecum were differentially enriched in FLUAV-SARS2 GSH. Within the feces, *Elusimicrobium* and unclassified Spriochaetaceae were differentially enriched in the SARS2 group while *Parabacteroides*, *Elusimicrobium*, *Christensenellaceae R-7 group*, *NK4A214 group*, *UCG-005*, and unclassified Spirochaetaceae were enriched in the FLUAV pre-exposed GSHs when compared to the mock. Unclassified Prevotellaceae, *Ligilactobacillus*, and unclassified Lachnospiraceae were differentially enriched in the mock. Compared to their respective pre-challenge feces, *Bifidobacterium*, *Allobaculum*, *Ileibacterium*, *Faecalibaculum*, and unclassified Eubacteriaceae were differentially enriched after SARS2 challenge, while an increase of *Prevotella* abundance was observed in the pre-challenge samples (S5 Table). No significant taxa were observed when comparing the FLUAV-SARS2 feces and their respective pre-challenge samples in at least 2/3 differential analysis (S5 Table). When comparing the two SARS2 challenge groups, *Anaerostipes* was enriched in the SARS2 GSHs while *Fibrobacter* and *UCG-005* were enriched in FLUAV-SARS2 GSHs. When analyzing the effect of FLUAV exposure prior to SARS2 challenge, *Anaerostipes* was enriched within the mock pre-exposed GSH (S5 Table). Overall, SARS2 respiratory infection in aged GSH with and without prior FLUAV exposure led to enhanced gut dysbiosis.

We further examined the association of genera and unclassified families to vRNA loads and pathology scores. Correlations among the heart and SI pathology were removed since the pathological score was the same for all GSH. We initially observed strong correlations among several bacterial taxa and the sample site (S9 Fig). However, when sample sites were analyzed separately, multiple genera, especially in the cecum and feces, had strong correlations among different infection factors, particularly with FLUAV pre-exposure and more severe pathology scores in the cecum (Fig 6). Taxa such as *Coxiella* in the SI, *UCG-005*, and *Fibrobacter* in the cecum and feces positively correlated with FLUAV exposure before SARS2 challenge, while *Peptococcus* in the cecum and *Lachnoclostridium* in the feces negatively correlated with FLUAV pre-exposure. Interestingly, *Bacteroidales RF16 group*, *Oscillibacter*, and multiple other genera within the SI, *Helicobacter* and *Christensenellaceae R-7* group within the cecum, and *NK4A214 group* within the feces positively correlated with more severe pathology in the cecum (Fig 6 and S1 Table). Meanwhile, *Bifidobacterium* in the SI, *Fournierella* and unclassified Bifidobacteriaceae and Christensenellaceae in the cecum, and unclassified Bifidobacteriaceae and *Allobaculum* in the feces negatively correlated with more severe pathology in the cecum. Other taxa were also strongly associated with other infection factors such as vRNA loads and histopathology in the respiratory tissues, particularly bacteria identified in the cecum. Overall, several bacteria were enriched among the different groups and correlated with vRNA loads and pathology, suggesting that intestinal microbes may play a role in disease pathology such as severity of lesions within the cecum during respiratory viral infection.

## Discussion

The present study uses an aged GSH model (14-months-old) to understand the pathobiology and respiratory/intestinal microbiota changes after SARS2 infection with and without previous FLUAV exposure. To our knowledge, there have been no previous studies that investigated the effect of SARS2 infection in 14-month aged GSH and how prior FLUAV exposure may affect the pathogenesis within this population subset. Investigating the interplay between FLUAV

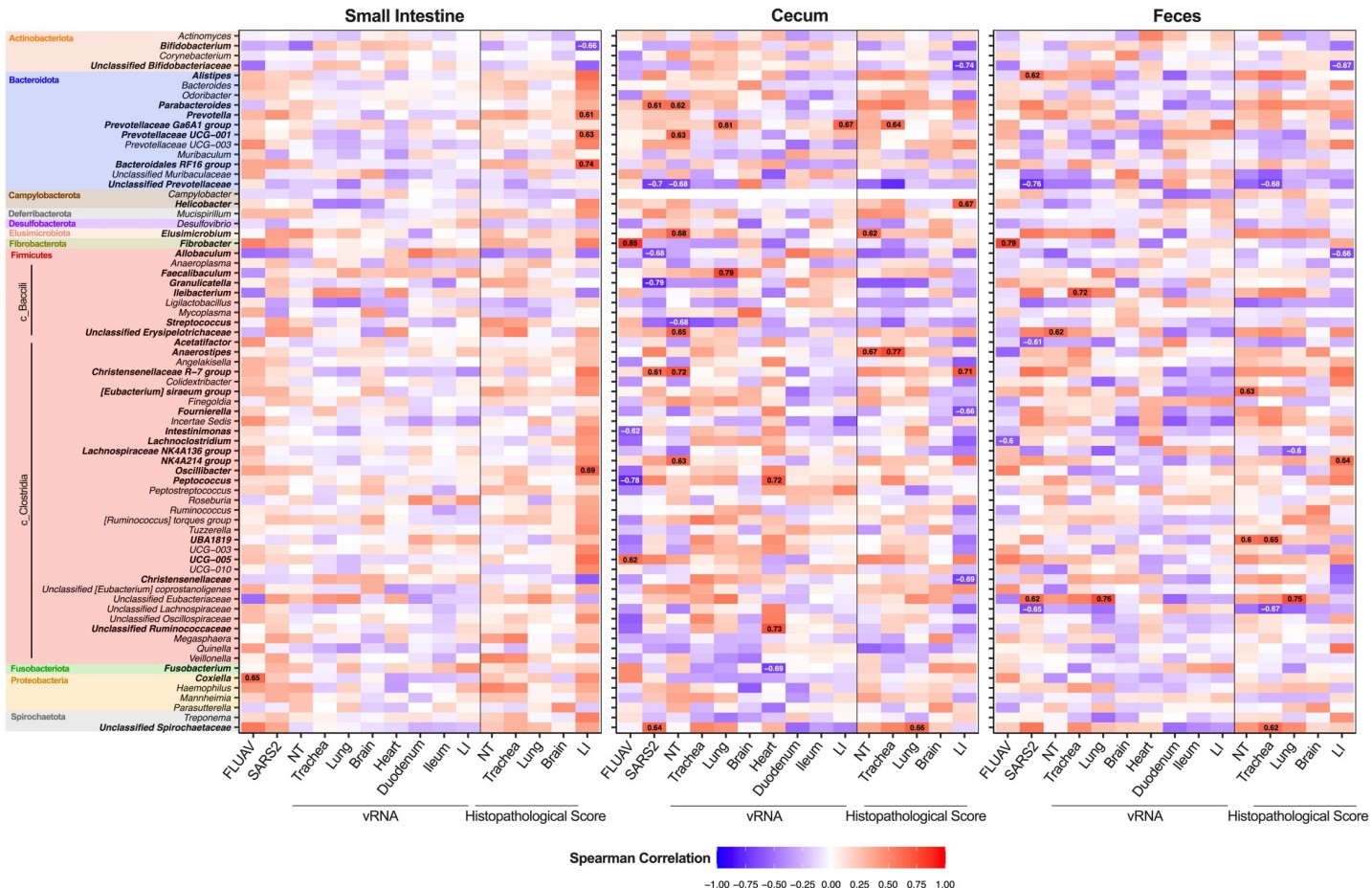

**Fig 6. Associations between taxa within the intestine and fecal microbiota and infection factors.** A heatmap illustrating the Spearman correlation among different infection factors and the relative abundance of bacterial taxa within the small intestine (duodenum and ileum), cecum and feces. Correlations among pathology of the heart and small intestine were removed since the pathological score was the same for all GSH. Taxa are color-coded by phylum. Correlation labels and taxa displayed in bold have a spearman correlation value > 0.60 or < -0.60. FLUAV–inoculated with FLUAV and then challenged with SARS-CoV-2 (mock-challenged GSH were not included), SARSCoV2 –challenged with SARS-CoV-2, NT- nasal turbinates.

pre-exposure and SARS2 in the elderly population is imperative as FLUAV H1N1 was the most common detected co-infecting virus with SARS2, and patients with viral co-infection had more severe symptoms and required ICU admission in a recent study [33].

Aging has been shown to affect the immune response leading to increased susceptibility to infections [34]. Understanding the effect of FLUAV pre-exposure to COVID-19 in the aging population is beneficial as seasonal influenza viruses are one of the most prevalent causes of morbidity and mortality in the elderly [35]. In this old-age GSH model utilized within this study, SARS2 infection led to weight loss starting at 2 dpc and high vRNA loads in the NT, trachea, lungs, and heart at 3 and 6 dpc. This is consistent with previous studies using aged GSH at 7–9 months showing peak weight loss at 6 to 7 dpc [12–14]. SARS2 and FLUAV-SARS2 GSHs had similar vRNA loads for respiratory and non-respiratory tissues except for significantly increased vRNA loads in the heart at 3 dpc and the NTs at 6 dpc in GSHs exposed to SARS2 only. Comparably, similar SARS2 vRNA loads within respiratory tissues between 6–8-weeks old GSHs infected with SARS2 and co-infected with FLUAV (PR8) H1N1 were observed at 4 dpc [8]. While we observed a reduction of vRNA in the NT at 6 dpc, a recent report showed that prior FLUAV infection 14 days before SARS2 infection resulted in reduced

SARS2 replication in the lung of male 6–8 weeks old GSH [36]. High SARS2 viral loads in respiratory samples, including NT, trachea, and lungs, at early time points, were also observed in 8–10-month-old GSH, consistent with our findings [12,13]. Here, pulmonary lesions were overall milder in SARS2 GSH than FLUAV-SARS2 GSH at 3 dpc, with moderate numbers of inflammatory cells, like what was previously observed in GSH co-infected with FLUAV [8]. However, more severe lesions were observed in SARS2 GSH compared to FLUAV-SARS2 GSH at 6 dpc. The dichotomy of these observations may be explained by faster disease progression and recovery in the FLUAV-SARS2 group compared to the SARS2 group or simply sample size effect.

Dysbiosis of the lung, intestine, and fecal microbiota composition was observed after SARS2 challenge with and without FLUAV pre-exposure. Numerous studies have reported that viral infections can affect the residential microbial composition and abundance within the respiratory tract (reviewed in [17]). In this study, beta diversity showed distinct clustering among both SARS2 challenged groups and the mock. Consistent with these observations, a previous study analyzing the oropharyngeal microbiota in humans showed beta diversity separations corresponding to COVID-19 severity [37]. In contrast, no statistically significant differences in alpha diversity in the lower respiratory tract were detected among SARS2 challenged GSHs and the mock controls, consistent with observations in the upper and lower respiratory tract of COVID19 patients [18,19,37].

The aged GSH lung microbiota was observed to be dominated by Firmicutes and Bacteroidetes at the phylum level and *Prevotella spp.*, *Veillona spp.*, and *Streptococcus spp.*, at the genus level, which is consistent with previous studies of healthy human lungs [17]. GSHs challenged with SARS2 exhibited an increased relative abundance of Proteobacteria at 3 and 6 dpc. The expansion of Proteobacteria members has previously been associated with FLUAV H1N1 infection and has been positively correlated with alveolar and systemic inflammation in patients with acute respiratory distress syndrome [17,38]. At the genus level, we observed enrichment of predominantly gram-negative bacteria within the genus *Prevotella 7*, *Campylobacter*, *Johnsonella*, *Fusobacterium*, *Haemophilus*, *Moraxella*, and gram-positive *Streptococcus* following SARS2 challenge. We also observed a positive correlation between *Actinomyces*, *Bifidobacterium*, *Prevotella 7*, and *Haemophilus* with more severe NT histological scores, while *Fusobacterium* positively correlated with more severe lung scores. *Streptococcus pneumoniae*, *Klebsiella pneumoniae*, and *Haemophilus influenzae* are reported among the most common bacterial co-infections in COVID-19 patients [39]. *Haemophilus* was also previously described as a hallmark of the oropharyngeal microbiome in critically ill COVID-19 patients [37]. Further, *Prevotella* spp. have been associated with increased inflammation by stimulating epithelial cells to produce IL-6, IL-8, and CCL20, which can then prompt recruitment of neutrophils and mucosal Th17 immune responses [40]. A recent study showed increased TGFβ, IL-17, and neutrophil recruitment within the respiratory tract of SARS2 infected aged GSH (40+ weeks) compared to young GSH (6–8 weeks) [14]. Similarly, moderate to large numbers of neutrophils, notably at 6 dpc, were also observed within this study by histopathology in both SARS2 groups. Elevated levels of neutrophils within the lung are a hallmark of COVID-19 and has been hypothesized to contribute to the severe disease pathology observed in aged individuals [14,41,42]. Increased serum levels of IL-6 and IL-8 have also been associated with disease severity in COVID-19 adult patients and enrichment of CCL20 has been observed in mechanically ventilated patients with COVID-19 [43–45]. In contrast, other studies showed that bronchoalveolar lavage (BAL) samples of 50–77 years old COVID-19 positive patients were associated with *Acinetobacter*, *Chryseobacterium*, *Burkholderia*, and Enterobacteriaceae, while COVID-19 negative pneumonia patients were associated with *Prevotella*, *Campylobacter*, *Streptococcus*, and *Haemophilus* [18,46]. The SARS2 challenged GSHs within this study

displayed more similar microbial signatures as observed in pneumonia patients as opposed to SARS2 patients, suggesting that bacteria within the genus *Prevotella*, *Campylobacter*, *Streptococcus*, and *Haemophilus* may be associated with a general respiratory disease pathology rather than a specific response to SARS2 infection.

Previous exposure to FLUAV in combination with SARS2 infection led to the enrichment of low abundance bacteria such as *Prevotella 7*, *Streptococcus*, *Johnsonella*, *Fusobacterium*, *Haemophilus*, and *Moraxella* within the lungs. *Haemophilus* was enriched in all GSHs at 3 and 6 dpc in the FLUAV-SARS2 group, while it was prevalent in only 1 out 3 GSHs at 6 dpc in the SARS2 group. These results suggest that previous innate immune response stimulation from FLUAV pre-exposure may influence the duration of opportunistic pathogen enrichment such as *Haemophilus* post-SARS2 infection. It has also been previously demonstrated that 6–8 week old GSH sustained heightened gene expression of interferon-stimulating genes 7 and 14 days post FLUAV infection [36]. The innate immune response, including macrophages and neutrophils, is important for viral clearance but also plays a role in regulating commensal microbes [47]. For instance, interferon-γ produced during T cell responses to FLUAV infection in mice correlated with suppression of phagocytosis by alveolar macrophages leading to the inhibition of initial bacterial clearance and the opportunity for secondary bacterial infections [48]. Respiratory viral infections can also promote bacterial colonization in the lower respiratory tract by excessive mucus production, decreased cilia movement, and impairment of the innate immune response leading to inefficient bacterial clearance resulting in the enrichment of opportunistic pathogens [47].

We also examined the response of the gut microbiota to infection. While previous studies in humans have detected vRNA in the intestine and feces correlating with occasional reports of gastrointestinal symptoms, we observed very low to no vRNA in the small or large intestine of GSH's, consistent with previous reports [22,26,49]. Numerous studies have also shown that acute and chronic lung diseases induce changes in gut microbiota through the gut-lung axis [23,50]. Due to the difficulty accessing intestinal samples, most human samples collected for microbiota analysis are fecal samples. Within human fecal samples, SARS2 infection has been associated with an increased relative abundance of opportunistic bacteria such as *Streptococcus*, *Rothia*, *Veillonella*, *Erysipelatoclostridium*, and *Actinomyces* while patients infected with FLUAV H1N1 was enriched in *Enterococcus*, *Prevotella*, *Finegoldia*, and *Peptoniphilus* [21]. Meanwhile, fecal samples from young male GSHs infected with SARS2 displayed increased abundance of *Oscillospiraceae UCG-007*, *Helicobacter*, *Escherichia-Shigella*, and *Parasuturella* and decreased abundance of short-chain fatty acid (SCFA)-producing Firmicutes such as *Ruminoccocus*, *Lachnospiraceae NK4A136*, *and Lachnospiraceae UCG-001* [50]. In accordance with previous studies, we observed enrichment of opportunistic bacteria including *Streptococcus* and *Helicobacter* in the SI and cecum while Lachnospiraceae was decreased in the feces of SARS2 challenged hamsters compared to the mock controls. Lachnospiraceae spp. have been previously described to produce SCFA, which plays a role in intestinal barrier integrity and modulation of pro-inflammatory factors, suggesting a beneficial role within a "healthy" gut microbiota [51]. In addition, increased relative abundance of *Mucispirillum*, Erysipelotrichaceae, and Spirochaetaceae (spirochetosis) was observed within cecum and feces of SARS2 challenged GSH with and without FLUAV exposure, in which spp. from these genera have been associated with inflammatory gastrointestinal disorders or diarrhea [52–54]. *Ruminococcaceae NK4A214 group*, *Helicobacter*, and *Chirstensenellaceae R-7* group within the cecum and feces also positively correlated with more severe pathology in the cecum and have been previously associated with chronic diseases such as chronic obstructive pulmonary disease, colorectal cancer, and ulcers [55–57].

Interestingly, multiple genera including *Bifidobacterium*, *Allobaculum*, *Ileibacterium*, *Faecalibaculum*, and unclassified Eubacteriaceae were enriched after SARS2 challenge compared to their respective pre-challenge fecal samples; however, no taxa were considered significantly

enriched in the FLUAV exposed GSHs when compared to their pre-challenge counterpart. The results suggest that initial stimulation of the innate immune response may result in greater changes of the intestinal microbiota compared to subsequent immune activations, or differences previously observed among the mock and FLUAV-SARS2 GSH can be explained by individual variability. The interplay between the gut microbiome and respiratory viral infections is bidirectional [47]. A respiratory virus infection can induce gut dysbiosis [47]; for instance, FLUAV infection favors type I interferon immune response which then promotes the outgrowth of Proteobacteria [58]. Contrastingly, the gut microbiome can also influence the immune response against the respiratory pathogen [47]; for example, macrophages from mice treated with antibiotics to induce depletion of the commensal bacteria exhibit decreased responses to type 1 and II interferons and reduced capacity to regulate viral replication during FLUAV infection [59].

Several limitations of this study must be noted: Despite the collection of multiple tissues for analysis, the study contained a relatively small sample size. And while the immune response plays a crucial role during early viral infection and the host microbiota, the immune response was not evaluated in this study. Future studies, including the analysis of the immune response, are needed to understand the effect of previous respiratory virus exposure to SARS2 infection and how the adaptive and innate immune response modulates the microbiota at the site of infection or through the gut-lung axis. We did not detect antibodies post-FLUAV H1N1 inoculation, which is similar to what has been previously observed in 2–3 month old GSHs [60]. While a recent study showed high viral titers of FLUAV H1N1 in GSHs, the study did not assess antibody titers for FLUAV [8]. Within this study, we did not evaluate FLUAV replication prior to SARS2 infection. However, FLUAV pre-exposure did have a distinct effect on the host microbial dynamics upon SARS2 infection compared to those that were only challenged with SARS2. Further research is needed to understand the impact of prior FLUAV infection and other respiratory pathogens and their role in stimulation of immune responses that might modulate outcomes in SARS2 infections due to effects on the residential respiratory and intestinal microbiota.

In summary, we characterized the pathogenicity and identified respiratory and gut microbiota changes during SARS2 infection with and without FLUAV pre-exposure in an animal model, emulating in this case, the elderly population. Information on the characterization of the lower respiratory tract in the elderly population is limited because of the difficulty of accessing samples from patients [23]. Because both SARS2 and FLUAV notably affect geriatric patients, understanding the disease pathology and the effect of the microbiota can potentially lead to the identification of differential host markers for disease severity within that demographic. While human BALF and fecal samples can be informative, human samples within elderly patients are more challenging to collect and tend to be highly variable and impacted by environmental conditions. As such, studying the effect of SARS2 infection on the microbiome in animal models in a controlled laboratory environment is a valuable alternative. In our work, specific microbial markers that have been previously reported in co-infections from opportunistic pathogens during COVID-19 were identified in the lungs of over aged GSHs challenged with SARS2 with and without FLUAV pre-exposure. Microbial changes within the different anatomical sections of the intestinal tract were also observed, warranting more in-depth studies investigating the complex relationship between the microbiota and pathogenicity in SARS2 in elderly patients.

## Materials and methods

### Ethics statement

Studies were approved by the Institutional Animal Care and Use Committee (IACUC) at the University of Georgia (Protocol A2019 03-032-Y2-A9). Studies were conducted under animal

biosafety level (ABSL) 2 and ABSL3 for SARS2 inoculation. Animal studies and procedures were performed according to the Institutional Animal Care and Use Committee Guidebook of the Office of Laboratory Animal Welfare and Public Health Service Policy on Humane Care and Use of Laboratory Animals. Animal studies were carried out in compliance with the Animal Research: Reporting of *In Vivo* Experiments (ARRIVE) guidelines (https://arriveguidelines.org). Animals were humanely euthanized following guidelines approved by the American Veterinary Medical Association (AVMA).

## Cells and virus

Virus stocks of A/Puerto Rico/08/1934 (H1N1) were generated in 10-day old specific pathogen-free (SPF) embryonated chicken eggs. Madin-Darby canine kidney (MDCK) cells were a kind gift from Robert Webster (St Jude Children's Research Hospital, Memphis, TN, USA) and used for FLUAV titration. Virus stocks were titrated by tissue culture infectious dose 50 ($TCID_{50}$), and virus titers were established by the Reed and Muench method [61]. Cells were maintained in Dulbecco's Modified Eagles Medium (DMEM, Sigma-Aldrich, St Louis, MO, USA) containing 10% fetal bovine serum (FBS, Sigma-Aldrich, St. Louis, MO, USA), 1% antibiotic/antimycotic (AB, Sigma-Aldrich, ST Louis, MO, USA) and 1% L-Glutamine (Sigma-Aldrich, St Louis, MO, USA). Cells were cultured at 37°C under 5% CO2.

Vero E6 Pasteur was kindly provided by Maria Pinto (Center for Virus research, University of Glasgow, Scotland, UK). Cells were maintained as previously described [62]. SARS2 (Isolate USA-WA1/2020) was kindly provided by Dr. S. Mark Tompkins (Department of Infectious Diseases, University of Georgia). Viral stocks were prepared in Vero E6 Pasteur cells. Briefly, cells in a T75 flask were incubated for 1 h with 1 ml of viral inoculum. After incubation, the inoculum was removed, and cells were cultured with DMEM containing 2% FBS 1% AB. After 96 h, the supernatant was collected, centrifuged at 15,000 g for 15 min, aliquoted, and stored at -80°C until use. Viral stocks were titrated by $TCID_{50}$ using the Reed and Muench method [61].

## Hamster studies

4-months-old female golden Syrian hamsters (GSH) were purchased from Charles River Laboratories (Kingston, NY) and housed individually in ABSL2 facilities at the University of Georgia. At 13-months of age, GSH were randomly distributed into three groups (Fig 1A), anesthetized with isoflurane, and challenged intranasally (i.n.) with 100 μl of either phosphate buffer saline (PBS) or $1x10^5$ TCID50/GSH of Influenza (FLUAV) A/Puerto Rico/08/1934 (H1N1). GSH were monitored daily for clinical signs of disease throughout the entire course of the experiment. Fourteen days later, GSHs were boosted in the same way as previously described. Twelve days after the boost, a subset of GSHs were bled through the cranial vena cava to confirm the absence of antibodies against SARS2, which was performed using the SARS2 surrogate virus neutralization test (Genscript, Piscataway, NJ). Thirteen days after the boost, GSHs were transferred to ABSL3 containment and then challenged intranasally (i.n.) with 100 μl of either PBS or $1x10^5$ TCID50/GSHs of SARS2 (Isolate USA-WA1/2020) the next day. At 3- and 6-days post-challenge (dpc), subsets of GSH were anesthetized with an intraperitoneal (i.p) injection (0.25–0.4 ml) of a ketamine/xylazine cocktail (150 mg/kg Ketamine, 10 mg/kg Xylazine), bled through the cranial vena cava, and then humanely euthanized with an intravenous injection of Euthasol (200mg/kg). Lungs, nasal turbinates (NT), trachea, heart, duodenum, ileum and cecum, and brain were collected from each GSH post-mortem. Tissues were stored at -80°C until further analysis. Cages were changed the day before euthanasia for fecal collection. Feces from individual cages were collected and immediately stored at -80°C until further processing.

## Tissue sample preparations

Tissue homogenates for virus titration were generated using the Tissue Lyzer II (Qiagen, Gaithersburg, MD). Briefly, 1ml of PBS-AB was added to each sample along with Tungsten carbide 3 mm beads (Qiagen). Samples were homogenized for 10 min and then centrifuged at 15,000 g for 10 min. Supernatants were collected, aliquoted, and stored at -80°C until further analysis.

## RNA extraction and RT-qPCR

RNAs were extracted from the tissue homogenates using the MagMax-96 AI/ND viral RNA isolation kit (ThermoFisher Scientific, Waltham, MA) following the manufacturer's protocol. A one-step real-time quantitative PCR (RT-qPCR) based on the Nucleoprotein gene segment was used as a surrogate of viral load, and it was employed using the primers 2019-nCov_N2-F (5'- TTACAAACATTGGCCGCAAA-3') and 2019-nCov_N2-R (5'- GCGCGACATTCCGA AGAA-3'). A probe with FAM as a reporter and TAMRA as a quencher was used (5'-FAM-ACAATTTGCCCCCAGCGCTTCAG-TAMRA-3'). The RT-qPCR was performed in a QuantStudio 3 Real-Time PCR System (ThermoFisher Scientific, Waltham, MA) using a Quantabio qScript XLT One-Step RT-qPCR ToughMix kit (Quantabio, Beverly, MA) in a 20 μl final reaction volume. Each reaction mixture contained a 1X master mix, 0.5 μM of each primer, 0.3 μM probe, and 5 μl of RNA. The qPCR cycling conditions were 50°C, 20 min; 95°C, 1 min, 40 cycles at 95°C, 1 min; 60°C, 1 min; and 72°C 1 s; with a final cooling step at 4°C. A standard curve was generated using 10-fold serial dilutions of a SARS2 virus stock of known titer to correlate RT-qPCR crossing point (Cp) values with the viral load from each tissue. Viral loads were calculated as $Log_{10}$ TCID50 equivalents/per gram of tissue.

## Histopathology and immunohistochemistry

Selected tissues included NT, trachea, lungs, heart, brain, and small and large intestines and were collected from each GSH in each group at 3- and 6-dpc for histopathological examination. Tissues were placed in 10% neutral-buffered formalin (NBF), fixed for at least 72 hours, paraffin-embedded, and processed for routine histopathology with hematoxylin and eosin staining (HE). A board-certified pathologist blinded to the study subjectively scored tissues based on percentage of the total parenchyma affected by lesions and inflammation as: none (0%), mild; <15% (1), mild to moderate; 16–30% (2), moderate; 31–50% (3), moderate to severe 51–75% (4) and severe; ≥75% (5). Features considered for the scoring were the following: presence and extent of necrosis, inflammation, endothelialitis, epithelial cell hypertrophy, hyperplasia, and regeneration (mitoses), mesothelial hyperplasia, and syncytia; presence and amount of intraluminal catarrhal/fibrino-necrotic/suppurative exudate, hemorrhage, edema, fibrin, and thrombi. Additionally, antibodies targeting the SARS-CoV-2 nucleocapsid (ThermoFisher Scientific, Waltham, MA; dilution 1/500), were also used to perform immunohistochemistry (IHC) on selected tissues.

## DNA extraction, amplicon library preparation, and sequencing

DNA was extracted by using a MoBio Power Soil kit (Qiagen, Gaithersburg, MD) with minor changes according to the earth microbiome protocol as follows: additional incubation at 65°C for 10 min after the addition of solution C1, beads were shaken at 20Hz for 20 min instead of 10 min, and samples were incubated at 4°C for 10 min instead of 5 min and then stored at -80°C until use. Following extraction, the V4 hypervariable region of the 16S rRNA gene was amplified using Phusion Hot Start 2 DNA polymerase (Thermo Fisher, Waltham, MA) using

primers 515F (59-GTGCCAGCMGCCGCGGTAA-39) and 806R (59-GGAC-TACHVGGGTWTCTAAT-39) in 20 μl PCR reaction (8.9 μl of molecular-grade water, 4 μl of 5X HF buffer, 0.4 μl of 10mM deoxynucleoside triphosphates [dNTPs], 1.25 μl of 10 mM 515F, 1.25 μl of 10 mM 806R, 4 μl of DNA, and 0.2 μl of polymerase) under the following conditions: 98˚C (30 s), followed by 25 cycles of 98˚C (10 s), 52˚C (30 s), and 72˚C (30 s), a final elongation step at 72˚C (5 min), and held at 4˚C. The PCRs were performed in duplicates and then visualized on a 1% agarose gel. Duplicate PCR products of the same sample were pooled in equal volumes, cleaned by 0.45X of Agencourt AMPure XP magnetic beads (Beckman Coulter, Pasadena, CA) according to the manufacturer's protocol, and eluted in molecular biology grade water (Genesee Scientific, San Diego, CA). Amplicon concentration was measured using the Qubit dsDNA HS assay kit (Thermo Fisher) on a Qubit 3.0 fluorometer (Thermo Fisher). DNA concentrations were normalized to 1.0 ng/ml. Subsequently, amplified DNA was used in a secondary amplification/dual barcode annealing reaction. Forward and reverse dual barcode primers (primers and barcodes with different reference indexes) were designed based upon primers generated by Caporaso et al. [63]. Secondary amplification reactions were performed using NEBNext high-fidelity 2X PCR master mix (NEB) in 50 μl reactions (26 μl of 2X mix, 20.5 μl of water, 2.5 μl of barcoded forward and reverse primers [10mM], 1 μl of DNA) under the following conditions: 98˚C (30 s), followed by four cycles of 98˚C (10 s), 52˚C (10 s), and 72˚C (10 s), followed by 6 cycles of 98˚C (10 s) and 72˚C (1 min), followed by a final extension of 72˚C (2 min) and then held at 4˚C. According to the manufacturer's protocol, samples were subsequently cleaned by 0.45X of Agencourt AMPure XP magnetic beads and eluted in molecular biology grade water. Fragment size distribution was analyzed on a subset of samples using the Agilent Bioanalyzer 2100 DNA-HS assay (Agilent, Santa Clara, CA, USA). Samples libraries were then normalized and pooled to a concentration of 2 nM based on a predicted total product size of 420 base pair (bp) using the Qubit dsDNA HS assay kit on the Qubit 3.0 fluorometer. The loading concentration of the pooled libraries was 8 pM. Libraries were sequenced using Illumina MiSeq V2 chemistry (Illumina, San Diego, CA), 2 by 250 paired-end. Negative controls, including an extraction blank and a PCR blank, were included in each sequencing run (2 runs total).

## Data analysis

Primer removal and demultiplexing were performed using Illumina BaseSpace using default settings. Sequence analysis was performed in R (version 4.1.0) with open-source software package DADA2 (version 1.16.0) [64,65]. Each sequencing batch was processed separately until chimera removal. For each batch, the quality of the raw paired-end reads was visualized and used to determine the appropriate truncation of read 1 (R1) by 10 bp and read 2 (R2) by 50 bp. After truncation, reads were discarded if they contained more than 2 maxEE "expected errors" or a quality score of less than or equal to 2. Following, each quality-filtered and trimmed read was processed independently by applying the trained DADA2 algorithm. The reads were then merged with a minimum overlap of 20 bp. After merging, both sequencing batches were combined, and chimeras were removed using the consensus method with default settings. Taxonomy was assigned in DADA2 using the native implementation of the naive Bayesian classifier using the Silva v.38 database. A count table and taxonomy file were created and used for downstream analysis. Lungs and intestine / fecal samples were analyzed separately. In the lungs, a total of 843,402 reads with an average of 52,712 reads/sample were generated. For the intestine/fecal samples, a total of 5,203,820 reads with an average of 58,469 reads/sample were generated. Samples were then filtered to remove amplicon sequence variants (ASV) with non-classified phylum or classified as Chloroplast or Mitochondria using RStudio

(version 2022.02.0) [66]. Further, ASV found in less than 5% of samples and had a total abundance of fewer than 10 reads were removed to avoid potential contaminants in diversity estimates. Post-filtering, an average of 49,290 reads / sample were generated for lung samples, 60,329 reads / sample for duodenum samples, 68,703 reads / sample for ileum samples, 52,636 reads / sample for cecum samples, and 49,809 reads / sample for fecal samples. Negative controls, including an extraction blank and a PCR blank, resulted in an average of 2 reads/sample (S10A Fig).

## Bioinformatic analysis

Alpha diversity, examined by the observed number of ASV and Shannon diversity, was calculated using rarified counts using phyloseq (version 1.38.0) [67]. Briefly, lung samples were rarified to 13,306 reads, and intestine/feces samples were rarified to 27,020 reads using command *rrarefy* in the vegan package (version 2.5.7) [68]. Rarefaction plots were generated using phyloseq.extended package (version 0.1.1.6) and edited using ggplot2 package (version 3.3.5) [69]. Rarefaction curves showed a similar number of observed species among the reads, suggesting adequate coverage (S10B Fig). The rarified counts were then imported into phyloseq, and diversity indexes were calculated using command *estimate richness*. Results were graphed using ggpubr package (version 0.4.0) [70]. Statistical comparison across groups was performed using the Kruskal-Wallis, while the Wilcox rank test with Bonferroni correction was used for pair-wise comparisons.

Regarding beta diversity, a non-metric multidimensional scaling (NMDS) plot showing unweighted Jaccard and weighted Bray-Curtis dissimilarity metrics were calculated with 3 dimensions, a maximum of 500 random starts and 999 maximum iterations using the rarified count data. NMDS plots and ellipses based upon the standard deviation were illustrated using *ordiellipse* from the vegan package. Beta diversity distances displayed in boxplots for comparison of within and across group distance metrics were created by calculating distances in vegan and then plotting using the ggplot2 package. Statistical pair-wise comparisons among groups were performed using the Wilcox rank test with Bonferroni correction. Multivariate statistics analysis, including permutational multivariate analysis of variance (PERMANOVA), were calculated using beta diversity distances with 1,000 permutations using vegan.

Relative abundances at the phylum and lowest classified level were generated using phyloseq. For the phylum taxonomic barplot, ASVs were agglomerated to the phylum level and then transformed into relative abundances using phyloseq. For the taxonomic barplot incorporating the lowest classification, relative abundances were calculated using phyloseq, and taxa with an overall prevalence of less than 1% were grouped. Differential analysis among groups of taxa at the genus level was performed using DeSeq2 (version 1.34.0) (S4A, S4C, S4E and S8A–S8C Figs), ALDEx2 (version 1.26.0) (S4B, S4D, S4F and S8A–S8C Figs), and linear discriminant analysis effect size (LEfSe) (Figs 4A–4B and S8A–S8C) [71–74]. ASVs in less than 20% of samples with a total abundance of fewer than 10 reads were removed for all differential analyses. For Deseq2 analysis, counts with an added value of +1, used to account for 0, were inputted, and a p-value of $< 0.01$ from the Wald test with Benjamini-Hochberg correction was considered significant. For ALDEx2 analysis, counts with an added value of +1 were inputted, and a p-value of $< 0.05$ from Welch's t-test with Benjamini-Hochberg correction was considered significant. For LEfSE analysis, rarified counts were inputted, and a p-value $< 0.05$ was considered significant. The p-value threshold used to determine significance for results produced in Deseq2 was more strict because of the high standard error than ALDEx2 and LEfSE. Relative abundances were also visualized using a heatmap of the differential taxa using ggplot2. Correlation analysis was performed to understand better the relationship between taxa and different

viral infection factors. Spearman correlation analysis was calculated by performing a log10 + 1 transformation on filtered count data and then calculated using the function *associate* from the microbiome package. The final plots were combined using Prism (v 9.1.0). Scripts used for analysis can be found on github at https://github.com/brittanyaseibert/Seibertetal_ SARS2FLUAV_AgedGSH.

## Graphs/Statistical analyses

Data analyses and graphs were performed using GraphPad Prism software version 9.3 (Graph-Pad Software Inc., San Diego, CA). Graphs were edited using PDF Expert (Readdle Inc, Odesa, Ukraine) and Affinity Photo (Serif, West Bridgford, United Kingdom).

## Supporting information

**S1 Fig. Immunohistochemistry staining in respiratory tissues of SARS2, FLUAV-SARS2, and mock-challenged hamsters at given timepoints.** (A-F) Nasal turbinates. (G-L) Lungs. (A-B) Variable amounts of intracytoplasmic virus antigen (red) immunolabeling is present in epithelial cells in challenged GSH. (D-E) Low levels of virus antigen are present in the epithelial cells as evidenced by faint red immunostaining. (C, F) No significant virus antigens are observed in mock hamsters' nasal turbinates. (G-H) Moderate amounts of virus antigen (red) are present in bronchiolar epithelium and intraluminal necrotic cellular debris in challenged GSH. (J-K) The intracytoplasmic virus antigen (red) is scarce at this timepoint. (I, L) No significant amount of virus antigen (red) immunolabeling are observed in the mock hamsters' lungs. All immunohistochemistry images are at 40X magnification.
(TIF)

**S2 Fig. Histopathologic findings in intestinal tissues of SARS2, FLUAV-SARS2, and mock-challenged hamsters at given timepoints.** (A-F) Small numbers of eosinophils, lymphocytes and macrophages are expanding the villi lamina propria (arrowheads) regardless of the timepoint and treatment groups. In the infected groups, the lacteals are dilated by small amounts of eosinophilic fluid (arrows). Increased numbers of protozoa compatible with Giardia spp. (insert) are populating the lumina in most sections. (G-H) At 3 dpc, moderate numbers of eosinophils, macrophages, and lymphocytes (arrowheads) are expanding the mucosa lamina propria in infected hamsters. The surface enterocytes are necrotic and sloughing into the lumen (arrow). The lumina contains mixed bacteria (insert) and protozoa. (I) In the mock infected hamsters, only small clusters of eosinophils are infiltrating the mucosa (arrowhead). Rare round protozoa (insert) are compatible with Entamoeba spp. (J and L) Small numbers of neutrophils infiltrate the mucosa lamina propria (arrowhead) in SARS2-only infected hamsters and mocks. Intestinal lumina contain mixed bacteria and a few round protozoa (insert) compatible with Entamoeba spp. (K) In FLUAV-SARS2 hamsters at 6 dpc, lesions recapitulate what is described at 3 dpc. The mucosa is infiltrated by mixed inflammatory cells (arrowheads), the glands lumen is dilated by increased bacteria (arrow), and enterocytes are sloughing into the lumen. All H&E images are pictured at 20X and inserts are at 40X magnification.
(TIF)

**S3 Fig. Beta diversity of the lungs in SARS2, FLUAV-SARS2 and mock-challenged GSH.** (A) NMDS plot of weighted Bray-Curtis dissimilarity distance of the lung samples. The color designates groups (SARS2: red, FLUAV-SARS2: blue, and mock: white) and dpc by the shape (3 dpc, circle and 6 dpc: square). Ellipses were constructed to include all points within the group. (B) Comparison of weighted Bray-Curtis dissimilarity distance within each group and across multiple comparisons of the lung samples. Intra-group variation is marked grey

shading, while the inter-group comparisons are marked gold. Pair-wise comparisons were performed using the Wilcox test with Bonferroni correction.
(TIF)

**S4 Fig. Differential taxonomic analysis of the lung microbiota using Deseq2, ALDEx2, and LefSE.** (A, C, E) Bacterial taxa identified at different relative abundances between (A) SARS2 (red), and mock (grey) or (C) FLUAV-SARS2 (blue), and mock (grey) or (E) SARS2 (red) and FLUAV-SARS2 (blue) groups using DeSeq2. Significantly enriched taxa were plotted with log2 fold change and the corresponding standard error estimate. (B, D, F) Effect size plot showing the median log2 fold difference (difference between) by the median log2 dispersion (difference within) when comparing (B) SARS2 and mock or (D) FLUAV-SARS2 and mock or (F) SARS2 and FLUAV-SARS2. Taxa considered significant by the Wilcox test are shown in green, taxa with BH corrected p-values are shown in red, while taxa with an effect size greater than 1.5 are outlined in blue. (G) Cladogram showing significantly ($p<0.05$) abundant taxa between SARS2 (red) and FLUAV-SARS2 (blue) using linear discriminant analysis effect size (LEfSe) analysis with default parameters. Taxa are color-coded by phylum, and bold taxa are considered significant in at least 2 of the 3 differential analyses performed (Deseq2, ALDEx2, and LefSE). Significant taxa using Deseq2 were determined by having a p-value < 0.01, while significant taxa using ALDEx2 and LEfSE were determined by having a p-value < 0.05.
(TIF)

**S5 Fig. Diversity analysis of the intestine and feces in SARS2, FLUAV-SARS2 and mock-challenged GSH.** (A) Shannon diversity of the intestinal/fecal samples between FLUAV-SARS2 (red), SARS2 (blue), mock (white), and pre-challenge (grey) containing all dpc from the rarified ASV count table. Multiple group comparisons were performed using Kruskal-Wallis, while pair-wise comparisons were conducted using the Wilcox test with Bonferroni correction. (B) Alpha diversity measure of the observed number of ASVs (left) and Shannon diversity (right) between FLUAV exposed (green) and mock (purple) pre-challenge fecal samples from the rarified ASV count table. Pair-wise comparisons were conducted using the Wilcox test with Bonferroni correction.
(TIF)

**S6 Fig. NMDS plots of the intestine and feces in SARS2, FLUAV-SARS2 and mock-challenged GSH.** NMDS plot of unweighted Jaccard distance (A, C, E, G) and weighted Bray-Curtis (B, C, F, H) dissimilarity distance of the small intestine, including the duodenum and ileum (A and B), cecum (C and D) and the feces (E and F). Groups are designated by the color (FLUAV-SARS2: red, SARS2: blue, mock: white, pre-challenge: grey) and sample type by the shape (duodenum: circle and ileum: square) or dpc by the shape (3 dpc, circle and 6 dpc: square). Ellipses were constructed using the standard deviation. Comparison between FLUAV exposure and mock controls when analyzing pre-challenge fecal samples were also compared (G and H).
(TIF)

**S7 Fig. Beta diversity and taxonomic abundances of the intestine and feces in SARS2, FLUAV-SARS2 and mock-challenged GSH.** Comparison of unweighted Jaccard dissimilarity distance within each group and across multiple comparisons in the small intestine, including the duodenum and ileum (A) and the cecum (B). Intra-group variation is marked with grey shading, while the inter-group comparisons are colored gold. Pair-wise comparisons were performed using the Wilcox test with Bonferroni correction. (C) Relative abundances agglomerated at the phylum level separated by sample type and group. (D) Relative abundances of each hamster at the lowest taxonomic rank identified separated by sample type, group, and ordered

by dpc.
(TIF)

**S8 Fig. Differential taxonomic analysis of the intestine and feces microbiota using Deseq2, ALDEx2, and LefSE.** Bacterial taxa identified at different relative abundances between (A left) SARS2 (red), and mock (grey) or (B left) FLUAV-SARS2 (blue), and mock (grey) or (C left) SARS2 (red) and FLUAV-SARS2 (blue) groups using DeSeq2 when all sample types (duodenum, ileum, cecum, and feces) were combined. Significantly enriched taxa were plotted with log2 fold change and the corresponding standard error estimate. Taxa in bold were those considered significant in at least 2 of the 3 differential analyses performed. Effect size plot showing the median log2 fold difference (difference between) by the median log2 dispersion (difference within) when comparing (A middle) SARS2, and mock or (B middle) FLUAV-SARS2, and mock or (C middle) SARS2 and FLUAV-SARS2 when all sample types (duodenum, ileum, cecum, and feces) were combined. Taxa considered significant by the Wilcox test are shown in green, taxa considered significant by BH corrected p-values are shown in red, while taxa with an effect size greater than 1.5 are outlined in blue. Cladogram showing significantly (p<0.05) abundant taxa between (A right) SARS2 (red) and mock (grey) or (B right) FLUAV-SARS2 (blue), and mock (grey) or (C right) SARS2 (red) and FLUAV-SARS2 (blue) using linear discriminant analysis effect size (LEfSe) analysis with default parameters when all sample types (duodenum, ileum, cecum, and feces) were combined. Taxa are color-coded by phylum, and bold taxa are considered significant in at least 2 of the 3 differential analyses performed. Significant taxa using Deseq2 were determined by having a p-value < 0.01, while significant taxa using ALDEx2 and LEfSE were determined by having a p-value < 0.05.
(TIF)

**S9 Fig. Associations between taxa within the intestine/fecal microbiota and infection factors.** A heatmap illustrating the Spearman correlation among different infection factors and the relative abundance of bacterial taxa when all sample types (duodenum, ileum, cecum, and feces) were combined. Correlations with histopathological score in the heart and small intestine were not included. Pre-challenge feces were also not included in this analysis. Taxa are color-coded by phylum. Correlation labels have a spearman correlation value > 0.60 or < -0.60. FLUAV–inoculated with FLUAV and then challenged with SARS-CoV-2 (mock-challenge GSH were not included), SARSCoV2 –challenged with SARS-CoV-2, NT- nasal turbinates, LI–large intestine.
(TIF)

**S10 Fig. Sample coverage.** (A) Sample coverage measured by the number of reads per sample separated by controls (DNA extraction and PCR blanks) and sample. The samples are color-coded by sample type (Lung: blue, Duodenum: pink, Ileum: yellow, Cecum: red, and Feces: orange). (B) Rarefaction curve of the lung samples colored by dpc (3 dpc: pink, 6 dpc: green). The dashed line (x = 13,306) represents the number of reads each sample was rarified for downstream diversity analyses.
(TIF)

**S1 Table. H&E pathology scores of different tissues.** Subjectively scored tissues (blind) based on percentage of the total parenchyma affected by lesions and inflammation as: none (0%), mild; <15% (1), mild to moderate; 16–30% (2), moderate; 31–50% (3), moderate to severe 51–75% (4) and severe; ≥75% (5). A dash is used to separate the score evaluated for each individual hamster separated by group and days post challenge (dpc).
(DOCX)

**S2 Table. Taxa considered significant among multiple differential analyses within the lungs.** Shown in bold are enriched groups that were considered significant in at least 2 of the 3 differential analyses performed. Adjusted p-values are reported: Deseq2 (Benjamini-Hochberg adjusted p-value) and ALDEx2 (Benjamini-Hochberg adjusted p-value using Wilcox t-test). p < 0.05 was considered significant for ALDEx2 and LefSE analysis and p < 0.01 was considered significant for Deseq2 analysis.
(DOCX)

**S3 Table. Taxa considered significant among multiple differential analyses of SARS-CoV-2 challenged aged Golden Syrian hamsters when samples from the intestine and feces were combined.** Pre-Challenge group was not included within these analyses. Shown in bold are enriched groups that were considered significant in at least 2 of the 3 differential analyses performed. Adjusted p-values are reported: Deseq2 (Benjamini-Hochberg adjusted p-value) and ALDEx2 (Benjamini-Hochberg adjusted p-value using Wilcox t-test). p < 0.05 was considered significant for ALDEx2 and LefSE analysis and p < 0.01 was considered significant for Deseq2 analysis.
(DOCX)

**S4 Table. Taxa considered significant among multiple differential analyses within the different anatomical sites in the intestine and feces.** Shown in bold are enriched groups that were considered significant in at least 2 of the 3 differential analyses performed. Adjusted p-values are reported: Deseq2 (Benjamini-Hochberg adjusted p-value) and ALDEx2 (Benjamini-Hochberg adjusted p-value using Wilcox t-test). p < 0.05 was considered significant for ALDEx2 and LefSE analysis and p < 0.01 was considered significant for Deseq2 analysis.
(DOCX)

**S5 Table. Taxa considered significant among multiple differential analyses comparing FLUAV pre-exposure to the mock pre-challenge and post-challenged fecal samples.** Shown in bold are enriched groups that were considered significant in at least 2 of the 3 differential analyses performed. Adjusted p-values are reported: Deseq2 (Benjamini-Hochberg adjusted p-value) and ALDEx2 (Benjamini-Hochberg adjusted p-value using Wilcox t-test). p < 0.05 was considered significant for ALDEx2 and LefSE analysis and p < 0.01 was considered significant for Deseq2 analysis.
(DOCX)

## Acknowledgments

We thank Kristine R. Wilcox and all the personnel from the Life Sciences vivarium and the Animal Health Research Center at the University of Georgia. We thank the personnel from the Animal Health Research Center, University of Georgia. We are also grateful to the Histology laboratory personnel, College of Veterinary Medicine, University of Georgia. CJC, BS, and DRP designed the experiments. CJC, BS, SCG, LCG, LO, GG, and DSR performed the *in vivo* studies, sample collection, and sample processing. SC performed histopathology examination and edited the manuscript. BS, CJC, EO, and DRP analyzed the data and wrote the manuscript. All authors approved the final version of the manuscript.

## Author Contributions

**Conceptualization:** Brittany Seibert, C. Joaquín Cáceres, Daniel R. Perez.

**Data curation:** Brittany Seibert, Silvia Carnaccini.

**Formal analysis:** Brittany Seibert, C. Joaquín Cáceres, Silvia Carnaccini, Daniela S. Rajao, Elizabeth Ottesen, Daniel R. Perez.

**Funding acquisition:** Daniel R. Perez.

**Investigation:** Brittany Seibert, C. Joaquín Cáceres, Stivalis Cardenas-Garcia, L. Claire Gay, Elizabeth Ottesen, Daniel R. Perez.

**Methodology:** Brittany Seibert, C. Joaquín Cáceres, Silvia Carnaccini, Stivalis Cardenas-Garcia, Lucia Ortiz, Ginger Geiger, Elizabeth Ottesen, Daniel R. Perez.

**Project administration:** Daniela S. Rajao, Daniel R. Perez.

**Resources:** Silvia Carnaccini, Daniel R. Perez.

**Software:** Brittany Seibert.

**Supervision:** Silvia Carnaccini, Daniela S. Rajao, Daniel R. Perez.

**Validation:** Silvia Carnaccini, Daniela S. Rajao, Elizabeth Ottesen, Daniel R. Perez.

**Visualization:** Silvia Carnaccini.

**Writing – original draft:** Brittany Seibert, C. Joaquín Cáceres, Silvia Carnaccini.

**Writing – review & editing:** Brittany Seibert, C. Joaquín Cáceres, Silvia Carnaccini, Stivalis Cardenas-Garcia, L. Claire Gay, Lucia Ortiz, Ginger Geiger, Daniela S. Rajao, Elizabeth Ottesen, Daniel R. Perez.

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
