## [Decision Letter · Decision Letter 0]

26 Aug 2022

Dear Dr. Perez,

Thank you very much for submitting your manuscript "Pathobiology and dysbiosis of the respiratory and intestinal microbiota in 14 months old Golden Syrian hamsters infected with SARS-CoV-2" for consideration at PLOS Pathogens. As with all papers reviewed by the journal, your manuscript was reviewed by members of the editorial board and by several independent reviewers. The reviewers appreciated the attention to an important topic. Based on the reviews, we are likely to accept this manuscript for publication, providing that you modify the manuscript according to the review recommendations.

The reviewers highlighted points that could be clarified in the text. They also identified experiments where more information on the choice of experimental design is needed to support the authors' interpretations. Additional supporting data validating the methods may be appropriate to include, particularly on the susceptibility of hamsters to PR8 virus. Finally, please address reviewer 3's comment on the lack of a control, young hamster comparison group in the discussion, as we realize repeating experiments in younger hamsters would take a lot of time and effort.

Sincerely,

Ashley L. St. John

Associate Editor

PLOS Pathogens

Andrew Pekosz

Section Editor

PLOS Pathogens

Kasturi Haldar

Editor-in-Chief

PLOS Pathogens

orcid.org/0000-0001-5065-158X

Michael Malim

Editor-in-Chief

PLOS Pathogens

orcid.org/0000-0002-7699-2064

The reviewers highlighted points that could be clarified in the text. They also identified experiments where more information on the choice of experimental design is needed to support the authors' interpretations. Additional supporting data validating the methods may be appropriate to include, particularly on the susceptibility of hamsters to PR8 virus.

Reviewer Comments (if any, and for reference):

Reviewer's Responses to Questions

**Part I - Summary**

Reviewer #1: In this manuscript, the authors observed that SARS-CoV-2 infection of aged Syrian hamsters resulted in high viral titers in the lung, corresponding pathology, and weight loss. They also found that SARS-CoV-2 infection can impact the lung and gut microbiota composition in the aged hamsters.

The results presented in the manuscript are interesting and provide new information regarding the mechanism of aged-related pathogenicity of COVID-19.

Reviewer #2: Seibert used two different infection models (SARS-CoV-2 and influenza A followed by SARS-CoV-2) to study the effect of SARS-CoV-2 infection on microbiota in aged hamster. This study is interesting, but the authors might want to address the following concerns as indicated below.

Reviewer #3: In this submission by Seibert, et al, the authors study SARS-CoV2 infection in older (14 mo) GSH, with and without prior FLUAV exposure and, in particular, characterize the effect on the lung and gut microbiomes. This work builds off their prior work on the effect of SARS-CoV2 on microbiota in a murine system. The effect of prior FLUAV on SARS-CoV2 disease and dysbiosis was modest. However, the authors did demonstrate significant changes in the composition of both lung and gut microbiomes.

The study appears to have been rigorously performed and the manuscript is well-written with appropriate figures. In humans, SARS-CoV2 clearly causes higher morbidity and mortality in older populations and the role of the microbiome in modulating responses to infections is increasingly appreciated. FLUAV coinfection is a common in older COVID patients. Thus, the study presented here has a strong scientific basis. The 14 mo-old hamsters used here are quite a bit older than those used in previous studies of age effects on SARS-CoV2 outcome (e.g. Osterrieder, et al. and Oishi, et al.) and is a strength. The effect of FLUAV coinfection in this model is also relevant, although diminished a bit in novelty with the recent publication from the tenOever lab (Oishi, et al, J. Virol. 2022).

The study has some issues which lessen its impact. First, all GSH used were aged 14 mo, without a direct comparison with younger GSH. So the effect of age cannot be determined, especially for effects on microbiota which can be influenced by the housing and diets of the animals. Therefore, comparisons with similar studies of dysbiosis in SARS-CoV2-infected GSH (e.g. Sencio, et al.) cannot be made. Second, while the microbiotic analyses were comprehensive and well-performed, the authors can only draw associations with disease severity. To their credit, the authors are careful not to overstate their conclusions. However, the manuscript lacks any mechanistic insights on the link between dysbiosis and disease outcome. They do discuss several possibilities (e.g. alterations in cyto/chemokine levels) but do not test them here. Third, there was no FLUAV-only group so the effects of FLUAV alone on pathology or microbiota cannot be assessed.

Overall, a well-conducted study and a well-written manuscript that is, however, somewhat limited in its scientific impact and scope.

**Part II – Major Issues: Key Experiments Required for Acceptance**

Reviewer #1: Page 3, lines 94–95. The authors mention that “To analyze the pathobiology, host response and effects of SARS2 and pre-exposure of influenza A virus……………”. Nevertheless, they infected the 14-month-old hamsters with an influenza A virus strain (i.e., A/Puerto Rico/8/34 (H1N1)), which does not replicate in this animal model. In fact, they did not detect antibodies post-A/Puerto Rico/08/34 (H1N1) inoculation. A previous study indicates that Syrian hamsters are highly susceptible to influenza A/H1N1 2009 pandemic viruses (Iwatsuki-Horimoto et al., JV, 2018). Such influenza A virus strains that can replicate well in the respiratory tracts of Syrian hamsters should be used.

Reviewer #2: 1. No antibody could be detected in the treated mice. Can the authors confirm that the influenza challenge has any immunological impact in the hamsters? Is hamster susceptible to PR8?

2. It is unclear why the authors assume the findings observed from the FLUAV-SARS2 is entirely due to innate responses. Can the authors exclude the potential effects from T cell responses?

3. This manuscript is highly descriptive. Many of them are not really so important or directly related to the microbiota results (e.g. most histopathology results between SARS and FLUAV-SARS2 are very similar). The reviewer has difficulties to link their findings. This reviewer finds the summary from some supplementary tables are much easier to follow.

4. Following the same line, it will be more useful to elaborate their findings from the microbiota analysis to the observation related to disease severity, or even the clinical observation found in humans.

5. Do RT-PCR positive tissues represent viremia or infected cells. Any data from immunological staining?

6. Except the microbiota data from fecal sample collected at day -1 before SARS-CoV-2 infection, there is no data about the effect of FLUAV exposure on the diversity of microbiota. This might make the causal relationship between SARS-CoV-2 and microbiota less clear cut.

Reviewer #3: None

**Part III – Minor Issues: Editorial and Data Presentation Modifications**

Reviewer #1: 1. Page 3, lines 93–94. “we utilized an aged Syrian hamster model (14 months equivalent to ~60 human years)”. The authors need to cite reference(s) regarding correlation between hamster age and human age.

2. Fig. 2. The authors mention that “At 6-dpc, pneumonia is characterized by florid pneumocyte type II……. (Page 7, lines 189–190); However, up to 95% of the alveoli were lined by florid proliferation of cuboidal epithelial cells (pneumocytes type II), replacing the necrotic pneumocytes type I (Fig 2P-Q) (Page 8, lines 231–232)”. However, it is not clear whether the cells indicated by asterisks are type II pneumocytes to the reviewer. Higher magnification images of H&E staining of hamster’s organs should be included for a complete evaluation of the reader with regards to the extent of inflammation and histopathological changes represented at each time point.

3. Fig.2. The authors need to provide the information on the magnification of the H&E image in the figure legend.

4. Page 21. How the A/Puerto Rico/8/34 (H1N1) virus was propagated should be included in the Experimental Procedures section.

5. Page 5, line 125. “HAI and ELISA” should be spelled out on the first use.

Reviewer #2: 1. Line 125: Day 27 day the FIRST FLUVA dose?

2. Supplementary tables: Please standardize terms “IAV-SARS2” and “FLUAV-SARS2” Or these are different models?

3. SARS2 should be read as SARS-CoV-2. The former is a disease, not a a virus.

Reviewer #3: The authors refer to modeling “geriatric patients” but GSH used here are equivalent to 60 y/o humans, 15 years younger than the currently accepted definition of “geriatric”. Suggest referring instead to modeling “older patients”.

The recent paper on SARS-CoV2 – FLUAV coinfection in hamsters (Oishi, et al, J. Virol. 2022) needs to be referenced and acknowledged in the Discussion.

PLOS authors have the option to publish the peer review history of their article (what does this mean?). If published, this will include your full peer review and any attached files.

Reviewer #1: No

Reviewer #2: No

Reviewer #3: **Yes: **Charles A. Scanga

Figure Files:

Data Requirements:

Reproducibility:

References:

---

## [Editor Report · Decision Letter 1]

28 Sep 2022

Dear Dr. Perez,

Thank you very much for submitting your manuscript "Pathobiology and dysbiosis of the respiratory and intestinal microbiota in 14 months old Golden Syrian hamsters infected with SARS-CoV-2" for consideration at PLOS Pathogens. As with all papers reviewed by the journal, your manuscript was reviewed by members of the editorial board and by several independent reviewers. The reviewers appreciated the attention to an important topic. Based on the reviews, we are likely to accept this manuscript for publication, providing that you modify the manuscript according to the review recommendations.

Although most of the reviewers' comments were addressed with revision, the lack of confirmation of IAV infection and viral replication in this model remains a concern. We request that the authors describe the evidence for a productive IAV infection in their animals, directly state whether this infection was confirmed or not in the results section, and to discuss any limitations to that data as a potential confounder in the manuscript.

Sincerely,

Ashley L. St. John

Associate Editor

PLOS Pathogens

Andrew Pekosz

Section Editor

PLOS Pathogens

Kasturi Haldar

Editor-in-Chief

PLOS Pathogens

orcid.org/0000-0001-5065-158X

Michael Malim

Editor-in-Chief

PLOS Pathogens

orcid.org/0000-0002-7699-2064

Although most of the reviewers' comments were addressed with revision, the lack of confirmation of IAV infection and viral replication in this model remains a concern. The editorial team request that the authors describe the evidence for a productive IAV infection in their animals, directly state whether this infection was confirmed or not in the results section, and to discuss any limitations to that data as a potential confounder in the manuscript.

Figure Files:

Data Requirements:

Reproducibility:

References:

---

## [Editor Report · Decision Letter 2]

5 Oct 2022

Dear Dr. Perez,

We are pleased to inform you that your manuscript 'Pathobiology and dysbiosis of the respiratory and intestinal microbiota in 14 months old Golden Syrian hamsters infected with SARS-CoV-2' has been provisionally accepted for publication in PLOS Pathogens.

Best regards,

Ashley L. St. John

Associate Editor

PLOS Pathogens

Andrew Pekosz

Section Editor

PLOS Pathogens

Kasturi Haldar

Editor-in-Chief

PLOS Pathogens

orcid.org/0000-0001-5065-158X

Michael Malim

Editor-in-Chief

PLOS Pathogens

orcid.org/0000-0002-7699-2064
---

## [Editor Report · Acceptance letter]

17 Oct 2022

Dear Dr. Perez,

We are delighted to inform you that your manuscript, "Pathobiology and dysbiosis of the respiratory and intestinal microbiota in 14 months old Golden Syrian hamsters infected with SARS-CoV-2," has been formally accepted for publication in PLOS Pathogens.

Best regards,

Kasturi Haldar

Editor-in-Chief

PLOS Pathogens

orcid.org/0000-0001-5065-158X

Michael Malim

Editor-in-Chief

PLOS Pathogens

orcid.org/0000-0002-7699-2064